# The Neural Frontier of Future Medical Imaging: A Review of Deep Learning for Brain Tumor Detection

**DOI:** 10.3390/jimaging11010002

**Published:** 2024-12-24

**Authors:** Tarek Berghout

**Affiliations:** Laboratory of Automation and Manufacturing Engineering, Department of Industrial Engineering, Batna 2 University, Batna 05000, Algeria; t.berghout@univ-batna2.dz

**Keywords:** artificial intelligence, brain tumor, cancer, classification, deep learning, MRI medical images

## Abstract

Brain tumor detection is crucial in medical research due to high mortality rates and treatment challenges. Early and accurate diagnosis is vital for improving patient outcomes, however, traditional methods, such as manual Magnetic Resonance Imaging (MRI) analysis, are often time-consuming and error-prone. The rise of deep learning has led to advanced models for automated brain tumor feature extraction, segmentation, and classification. Despite these advancements, comprehensive reviews synthesizing recent findings remain scarce. By analyzing over 100 research papers over past half-decade (2019–2024), this review fills that gap, exploring the latest methods and paradigms, summarizing key concepts, challenges, datasets, and offering insights into future directions for brain tumor detection using deep learning. This review also incorporates an analysis of previous reviews and targets three main aspects: feature extraction, segmentation, and classification. The results revealed that research primarily focuses on Convolutional Neural Networks (CNNs) and their variants, with a strong emphasis on transfer learning using pre-trained models. Other methods, such as Generative Adversarial Networks (GANs) and Autoencoders, are used for feature extraction, while Recurrent Neural Networks (RNNs) are employed for time-sequence modeling. Some models integrate with Internet of Things (IoT) frameworks or federated learning for real-time diagnostics and privacy, often paired with optimization algorithms. However, the adoption of eXplainable AI (XAI) remains limited, despite its importance in building trust in medical diagnostics. Finally, this review outlines future opportunities, focusing on image quality, underexplored deep learning techniques, expanding datasets, and exploring deeper learning representations and model behavior such as recurrent expansion to advance medical imaging diagnostics.

## 1. Introduction

Brain tumors, both benign and malignant, are a major focus of medical research due to their impact on patient health and the need for better diagnostic and therapeutic strategies [1,2,3,4]. These tumors can be classified into primary brain tumors, such as gliomas, meningiomas, and pituitary tumors, and metastatic tumors that spread from other body cancers [5,6,7,8,9,10,11,12,13]. Gliomas, including glioblastomas, are the most aggressive, while meningiomas are typically benign and originate in the meninges [8,9,10,11,12,13]. Symptoms of brain tumors vary depending on size, location, and type, and can include headaches, dizziness, cognitive changes like forgetfulness, and physical impairments such as difficulty with mobility and speech. Emotional and behavioral changes, like mood swings and reduced social engagement, also occur [14]. Figure 1 illustrates these diverse symptoms, emphasizing the complex effects on both physical and psychological well-being [14].

The causes of brain tumors are not fully understood, but genetic mutations and environmental factors, such as radiation exposure, may play a role [8,9,10,11,12,13]. Diagnosis is typically performed using imaging techniques like MRI or CT scans to determine the tumor’s location and size [1]. Treatment often involves surgery, radiation therapy, and chemotherapy, tailored to the tumor type, location, and patient factors such as age and overall health. Surgery aims to remove as much of the tumor as possible, while radiation therapy and chemotherapy target and destroy tumor cells. Advances in treatment have improved outcomes, but prognosis varies based on tumor characteristics and early detection. Early diagnosis and treatment are critical for better survival rates and quality of life. Figure 2 summarizes the key challenges, causes, treatment options, and diagnostic methods for brain tumors, highlighting their significance in both research and clinical practice due to their high mortality rates and complex treatment [15,16,17,18,19,20,21,22,23,24].

Conventional diagnostic methods, such as manual analysis of MRI scans and histopathological examinations, are often time-consuming, subjective, and prone to human error. These limitations can significantly hinder timely and accurate diagnosis of brain tumors [25,26]. However, in recent years, deep learning has emerged as a leading technology in medical imaging. Deep learning offers promising solutions for automating critical tasks such as brain tumor feature extraction, segmentation, and classification, potentially improving both the speed and accuracy of diagnosis [27]. Leveraging advanced algorithms and vast datasets, deep learning models can enhance diagnostic accuracy and efficiency, providing critical support in clinical decision-making. Given the abundance of studies exploring deep learning applications in brain tumor detection, a comprehensive overview is essential to synthesize findings and identify research gaps [28,29,30,31,32,33,34,35,36].

This introduction is dedicated to three key subsections. The Section 1.1 will delve into the methodologies used in selecting and analyzing relevant papers, highlighting the criteria for analysis and the approaches employed to derive meaningful insights. The Section 1.2 will express the significant contributions of this review, emphasizing how it synthesizes existing research to identify gaps and propose future directions in brain tumor detection. Finally, the Section 1.3 will provide an overview of the paper’s structure, detailing the main sections and presenting a hierarchical classification of the methodologies and findings discussed throughout the review. Through this organized approach, we aim to clarify the critical role of deep learning in advancing brain tumor detection and address the pressing need for continued innovation in this field.

### 1.1. Paper Collections Methodologies and Analysis Criteria

The paper collection process follows a well-defined and organized methodology. Specific keywords have been established to ensure their presence in the metadata of the research papers, primarily including terms like “image processing”, “medical”, “deep learning”, “tumors”, “machine learning”, “diagnosis”, and “neural networks”. To enhance the accuracy of search results, the terms “tumor + brain” or “brain” are required to be present in the title. To streamline the search mechanism, a “Publish or Perish” software v8.16.4790.9060 is employed, targeting databases such as Scopus, PubMed, and Google Scholar search engine [37]. Research is conducted on an annual basis, starting from 1 January 2019 to 17 October 2024 (i.e., the day the development of this review paper began), with the aim of covering at least the past half-decade to draw significant conclusions. The search results are saved as “*.csv” files containing comprehensive details such as title, year, source, publisher, etc. Subsequently, a MATLAB script is developed to further clean the collected data and simplify the process of downloading the papers. It begins by filtering the collected dataset of those “*.csv” files to include only entries where the title contains the word “brain”, or “tumor”, irrespective of case. This step ensures that the analysis remains concentrated on relevant studies.

The filtered datasets are then concatenated into a single table, and duplicate entries based on the title are removed. The script also extracts the research types, enabling focused analysis on different categories of research papers. For each unique type, the script creates a separate table. This organization facilitates easier access to the data during subsequent analysis. The individual tables are sorted by publication year in descending order to maintain a consistent format for analysis. The combined dataset, containing all relevant tables, is saved as a “*.mat” file for future use. This step ensures that the data are stored in a structured format, allowing for efficient retrieval and further analysis in later sessions. In the subsequent part of the analysis, statistics regarding the number of research papers by type are calculated. A list of all the types and stores statistics are then generated, including the type name and the corresponding count of research papers.

Figure 3a visualizes the distribution of research papers by type, represented as a pie chart displaying the percentage distribution of various types of research papers. The chart indicates that the majority of the papers are categorized as articles, accounting for 62.8% of the total publications, which corresponds to 81 papers. This highlights a strong emphasis on original research within the field. In contrast, conference papers constitute 25.6% of the total, totaling 33 papers, suggesting significant interest in disseminating research findings through conferences. Book chapters account for a minor segment of the dataset at 3.1%, totaling four papers, possibly reflecting a more specialized or less common avenue for research dissemination.

Figure 3b visualizes the number of research papers published per year for each type, revealing notable trends in publication output across different research paper types from 2019 to 2024. For articles, there is a significant upward trajectory, starting with four papers in 2019, increasing to six papers in both 2020 and 2021, and then experiencing a substantial surge to 24 papers in 2022. This increase reflects a growing momentum in research activity, continuing with 16 papers published in 2023 and reaching a peak of 25 papers in 2024. This consistent rise suggests a strong focus on original research in recent years. In contrast, the publication of book chapters has been relatively limited, with no publications in 2019, followed by one paper in 2020, and then no further publications until 2022, when two papers were released. In 2023, only one paper was published, and there were no contributions in 2024. This indicates that book chapters are not a primary avenue for research dissemination in this field. Conference papers exhibit a gradual increase over the years, starting with one paper in 2019 and rising to two papers in 2020 and 2021. In 2022, the number increased to seven papers, followed by thirteen papers in 2023, and eight papers in 2024. This suggests that conference presentations have become a more popular method for sharing research findings, particularly in recent years.

According to Figure 3a,b, review articles make up 8.5% of the dataset, represented by 11 papers. This indicates that, while reviews are valuable for synthesizing existing research, they are less common compared to original studies and conference contributions. This trend underscores the need for more review articles in the field to provide comprehensive overviews and identify gaps for future research. In this analysis, we were able to download a total of 55 journal articles and 22 conference papers, which will serve as the primary focus of our study. These papers are divided into three key sections: extraction, segmentation, and classification, with approximately 27 papers dedicated to each section. Important conclusions from these papers will be collected and analyzed to provide a comprehensive overview of the current research landscape. Additionally, we downloaded nine review papers, which will be analyzed separately to identify research gaps and determine the limitations of previous review-based studies. These papers will contribute to a more thorough understanding of the challenges and advancements in the field.

It should be noted that the author did not intentionally exclude papers from any specific publisher during the paper collection process, which was based on the aforementioned criteria. As stated, the filtering process was carried out using a program script designed to minimize human error, such as overlooking certain papers or introducing biases due to lapses in focus. While it is possible that some articles were unintentionally excluded, it is guaranteed that the selected papers are directly relevant to the topic of deep learning and brain tumor detection. Although no methodology is perfect and some exclusions may have occurred, the quantity and quality of the selected papers are deemed sufficient to draw generalized conclusions.

In this work, at the preliminary stage, based solely on the titles of the selected papers, we can conclude that the research carried out in the field of brain tumor detection with topics of classification, feature extraction, and segmentation primarily explores a broad range of methods and paradigms. By “methods”, we refer to algorithms and models used to process medical images, including CNNs, LSTMs, Autoencoders, and GANs, alongside optimization techniques like Bayesian optimization for instance, to improve feature extraction and model accuracy, especially for MRI scans. Similarly, “paradigms” refer to frameworks guiding these methods, such as transfer learning for adapting pre-trained models, reinforcement learning for performance improvement, federated learning for privacy-preserving data usage, and XAI for making model decisions transparent and interpretable for medical professionals. It is noteworthy to mention that this is an author-specified concept that must be taken into consideration throughout the entire review.

In context of methods, the papers adopt advanced deep learning techniques, such as CNNs, LSTM, Gated Recurrent Units (GRUs), and autoencoders, with some leveraging of GANs for synthetic data generation. Several papers incorporate hybrid models combining various networks, such as in the case of hybrid deep learning models using CNNs alongside optimization algorithms like Bayesian optimization. Moreover, optimization techniques such as Particle Swarm Optimization (PSO), Genetic Algorithms (GA), and Ant-Lion Optimization (ALO) are frequently employed to enhance model performance or fine-tune hyperparameters. Other optimization and feature extraction techniques like Principal Component Analysis (PCA) and wavelet transforms are used to enhance brain tumor image data. Statistical tests are less emphasized compared to algorithmic-based solutions but are sometimes applied for validation and performance metrics.

In terms of learning paradigms, several papers explore transfer learning, which allows models to leverage pre-trained architectures, typically from large datasets like ImageNet, to fine-tune for specific tasks in brain tumor classification. Reinforcement learning and its application in medical imaging appear in certain studies, demonstrating its potential to improve model decision-making over time. Federated learning is another emerging approach, enabling the use of distributed data without compromising privacy, particularly in healthcare. There is also a notable focus on XAI and Interpretable AI (IAI), which aim to make deep learning models more transparent and understandable in high-stakes scenarios like medical diagnosis. Additionally, some papers integrate IoT solutions to facilitate real-time data acquisition and processing, improving the connectivity and application of artificial intelligence in clinical settings. Thus, the carried-out research not only focuses on leveraging cutting-edge algorithms and optimization techniques but also explores innovative learning paradigms such as transfer learning and XAI to ensure more robust and interpretable models.

This combination of methods and paradigms reflects the continued push toward enhancing the accuracy and practicality of brain tumor diagnosis through deep learning technologies. Table 1 introduces and summarizes the analysis criteria for brain tumor feature extraction, segmentation, and classification methods, detailing the various algorithms, models, and paradigms adopted in this work.

### 1.2. Contributions

This paper provides a comprehensive review of trends in brain tumors detection with deep learning by focusing on advancements in image-based feature extraction, segmentation, and classification, with a specific focus on the application of deep learning techniques across these stages. By analyzing methods, trends, and gaps in existing research, this paper aims to bridge critical knowledge gaps and foster innovation in the field of computer vision and clinical diagnosis. The key contributions of this paper are as follows:
**Exploration of research methodologies and trends:** This paper offers a detailed analysis of research methodologies and statistical trends in the field, providing a holistic view of the current scientific landscape (see Section 1.1).**Analysis of related review papers:** By critically examining related review articles, this study identifies prior contributions and uncovers existing gaps, justifying the need for this review and offering a roadmap for future research efforts in image processing using deep learning.**Comprehensive review of deep learning advancements:** The paper provides an in-depth overview of recent developments in deep learning techniques for feature extraction, segmentation, and classification, underscoring their effectiveness in automating complex tasks and improving performance in various applications.**Integration of feature extraction, segmentation, and classification:** This study emphasizes the interconnected nature of feature extraction, segmentation, and classification in the image analysis pipeline, demonstrating how their integration can enhance accuracy and reliability in a wide range of use cases.**Critical evaluation of algorithms and methodologies:** The paper reviews these modern approaches, analyzing the strengths, weaknesses, and practical implications of various algorithms used in image-based feature extraction and classification.**Exploration of datasets and their role in research:** An investigation into the datasets used for training and evaluation highlights the importance of diversity, quality, and preprocessing in ensuring robust and generalizable models.**Identification of future research directions:** The paper outlines pathways for future opportunities, such as image quality and compression, leveraging multimodal data, improving deep learning architectures, and addressing challenges in data accessibility and annotation.

### 1.3. Outline of Paper Sections

As illustrated in the simplified diagram in Figure 4, this review paper is structured to provide a comprehensive overview of the state of research in brain tumor detection using medical images. Following the introduction in Section 1, Section 2 identifies key gaps in existing literature by analyzing related review papers. The heart of the paper lies in Section 3, Section 4 and Section 5, where we delve into the latest trends and challenges in image-based feature extraction (Section 3), image segmentation (Section 4), and image classification (Section 5). These sections provide detailed overviews of the fundamental concepts, challenges, and advancements in each area, including techniques such as CNNs, GANs, Autoencoders, and optimization algorithms in a simplified way in a sort of illustrative examples. They also explore emerging paradigms like transfer learning, reinforcement learning, and the IoT as they pertain to medical imaging and tumor detection. Finally, Section 6 outlines promising directions for future research, while Section 7 concludes the paper by summarizing the key insights and contributions. This structured approach allows a holistic understanding of the current state of brain tumor detection research, while also paving the way for future advancements in the field.

In this architecture of the current review paper, it is acknowledged that the long paragraphs discussing the papers may be difficult to read. To address this, we have made efforts to improve readability by introducing subsections. However, we believe the current subsectioning is the most suitable in this case, as it allows for clear presentation of the review results in tables, followed by bar charts. Furthermore, the review is structured around three major axes: classification, extraction, and segmentation. We believe that the inclusion of bar charts and diagrams will greatly assist readers in understanding the recent trends in each subfield and provide valuable insights for drawing meaningful conclusions. We trust that this organization will enhance the clarity of the manuscript while ensuring a comprehensive review of the topic.

## 2. Related Review Papers: Exploring Gaps in Research

This section is dedicated to analyzing related review papers on brain tumor detection or classification, focusing on several key criteria to identify research gaps. Specifically, it evaluates what each review paper aimed to achieve, how the authors believe their work contributes to the field, and the advantages these reviews offer in terms of summarizing various methods. It also assesses the depth of coverage on deep learning approaches, examining whether these reviews explore classification, segmentation, and extraction tasks comprehensively or briefly. Additionally, this section evaluates whether these studies present and analyze, or at least scan, substantial datasets utilized in previous research. Research gaps and future opportunities identified in these reviews are highlighted, along with limitations, particularly in the context of deep learning, including the extent to which detection, segmentation, classification, and extraction are discussed, and whether the latest advancements in deep learning models and hybrid techniques are addressed.

The review research in [28] provides a comprehensive overview of studies for glioma brain tumors classification using MRI images and deep learning. Accordingly, 77 research papers between 2010 and 2022, describing different methods applied to glioma detection are analyzed while giving a focus to classification and segmentation. More specifically, the authors focus on how deep learning models, particularly CNNs and their wide-spread architectures like U-Net, have been employed for medical image segmentation and the use of transfer learning for classification tasks. The authors, from their perspective, believe that this review is valuable as it offers a global view of existing methods, and helps to identify areas where further investigation is needed, particularly in glioma detection and grading, areas that have received less attention compared to segmentation. Advantages of the paper include its clear focus on methodological aspects such as pre-processing, model architecture, and evaluation criteria, and its meta-analysis of results. Deep learning is explored in depth, with detailed discussions on its applications in MRI image analysis for brain tumor segmentation, classification, and detection. Key models like CNNs, U-Net, and transfer learning approaches are covered, showcasing the advantages of deep learning’s automatic feature extraction over traditional machine learning methods. The review explored several major public datasets, including the Brain Tumor Segmentation (BraTS) [38,39,40] dataset, which was reviewed in 31 articles, as well as the TCIA and other datasets like Information eXtraction from Images (IXI) [41] and Ischemic Stroke Lesion Segmentation (ISLES) [42] datasets. While the review provides significant insights, the authors also identified research gaps, noting that while segmentation has been extensively studied, topics such as glioma classification and feature extraction have not received as much attention. Additionally, the paper highlights certain limitations, including the predominant focus on a single type of brain tumor, while other types, such as meningioma tumors and pituitary tumors, are equally important and warrant further exploration. In addition, there is a limited exploration of advanced deep learning models like GANs which could enhance data augmentation and improve model robustness. Other methods, like transformer-based architectures and hybrid models combining CNNs with recurrent networks (e.g., LSTMs), also remain underutilized in this domain. In ref. [29], the authors conducted a review on brain tumor segmentation using MRI, analyzing 79 papers from 2019 to 2023. They categorized used methods into non-artificial intelligence, machine learning, deep learning, and hybrid approaches, aiming to provide an overview of current techniques, their achievements, limitations, and challenges hindering clinical adoption. It discusses deep learning architectures like CNNs, U-Net, transformers, GANs, and hybrids, highlighting their performance and limitations. The authors emphasize the BraTS dataset [38,39,40], used in 84% of studies, and briefly mention less frequently used datasets like TCIA. The paper highlights research gaps, such as the challenges of generalizing models across different MRI scanners, sensitivity to noise, performance inconsistency in segmenting tumor subregions, and the need for efficient models to balance accuracy and computational costs. The authors emphasize that further research is needed to overcome these limitations to make them viable for real-world clinical applications. From the deep learning perspective, limitations discussed include high computational costs, challenges related to data imbalances, noise variability in MRI scans, and difficulties in subregion segmentation due to complex tumor shapes and poor contrast. Although segmentation was the primary focus, some classification aspects were also discussed, with the recognition that improving segmentation accuracy, particularly for difficult-to-identify tumor subregions, remains a significant research gap. In ref. [30], the authors investigated machine learning and deep learning algorithms for brain tumor detection, focusing on segmentation and classification in the context of MRI imaging. They evaluated various techniques, highlighting their strengths, limitations, and notable successes, while identifying research gaps and potential areas for future exploration. The review aims to serve as a valuable resource by providing a systematic breakdown of methods, performance metrics, and methodologies to improve tumor diagnosis. Deep learning is explored in detail, particularly the use of CNNs for segmentation and classification tasks. The authors emphasize the importance of datasets, referencing the BraTS dataset [38,39,40], widely used for evaluating deep learning models, and identify opportunities to enhance glioma detection precision and segmentation techniques for larger datasets. They also address challenges in accurately segmenting tumor regions, including whole, core, and enhancing tumors. However, feature extraction received less emphasis in their discussion, indicating an opportunity to explore its potential further in improving model performance and diagnostic accuracy. In ref. [31], the authors reviewed various image segmentation methods, focusing on MRI-based brain tumor detection using machine learning. They compared traditional and machine learning models, emphasizing deep learning approaches. The review covered research from 1998 to 2020, analyzing segmentation algorithms and their accuracy in detecting brain tumors. It highlights key advancements and identifies gaps, guiding future research in brain tumor detection. The review’s strengths include comprehensively covering two decades of research, comparing traditional and modern methods, and emphasizing deep learning’s superior performance. It also discusses challenges such as MRI noise, data handling, and tumor/non-tumor imbalance. Deep learning is explored with models like CNNs and U-Net, which are highlighted for their accuracy in handling complex image features. The paper analyzes these models’ applications, challenges, and future potential. Additionally, the authors discuss datasets used in brain tumor segmentation, particularly the BraTS [38,39,40] challenge datasets. A limitation of the review is its focus on segmentation, with less emphasis on feature extraction and classification. This leaves a gap in understanding how feature engineering could complement segmentation techniques to enhance model robustness and diagnostic accuracy, particularly in scenarios requiring complex differentiation between tumor subtypes or stages. In ref. [32], the authors reviewed machine learning and deep learning techniques for brain tumor analysis in medical imaging. They summarized approaches for feature extraction, segmentation, and classification, highlighting the performance of models such as CNNs. The review discusses challenges and opportunities for improving these techniques, outlining the strengths and limitations of various algorithms and clarifying trade-offs between accuracy, complexity, and efficiency. The authors also reviewed MRI brain scan datasets used to train and validate the discussed models, focusing on tumors such as pituitary, glioma, and meningioma. Notable research gaps include the need for better integration of low- and high-level features. The review suggests opportunities for developing models that learn significant features and integrate CNNs with other classifiers to achieve improved accuracy. However, limitations of the review include a primary focus on CNNs and classification tasks, with less attention to the latest advances in learning paradigms, such as transfer learning and GANs for instance, and its implications for the field. In ref. [33], the authors evaluate various techniques for brain tumor segmentation using MRI data. The review covers thresholding, region-based methods, clustering techniques, and hybrid approaches, providing insights into both supervised and unsupervised methods. Its strength lies in the comprehensive coverage of techniques, from classical methods to advanced machine learning approaches, including a detailed analysis of deep learning models like CNNs. The authors also emphasize the importance of large datasets, such as BraTS [38,39,40] and the Internet Brain Segmentation Repository (IBSR) [43], for benchmarking segmentation methods. Research gaps include limited focus on hybrid techniques, unsupervised learning, real-time segmentation (e.g., Autoencoders), and advanced architectures like GANs. Additionally, the review prioritizes segmentation over classification, detection, and end-to-end systems integration. In ref. [34], the authors presented a review of deep learning techniques applied to MRI brain tumor analysis, focusing on 92 papers from 2018 to 2020 across different databases. They concentrated on two primary tasks: classification and segmentation of brain tumors using deep learning, particularly CNNs. A key strength of the review is its structured analysis, offering clarity on the strengths and limitations of various models, and its discussion of commonly used datasets such as BraTS [38,39,40] and the Cancer Imaging Archive (TCIA) datasets like The Cancer Genome Atlas Glioblastoma Multiforme (TCGA-GBM) [44] and The Cancer Genome Atlas Low Grade Glioma (TCGA-LGG) [45]. The review explores CNN architectures and pretrained models (U-Net, VGG, AlexNet, ResNet), their applications, technical layers, and performance metrics. It also covers advanced topics such as transfer learning, data augmentation, and challenges like black-box modeling. The review’s gaps include the need for discussions on model interpretability and debate solutions for small datasets. Additionally, the review heavily focuses on segmentation with CNNs, while classification and feature extraction receive less attention. In ref. [35], the authors reviewed artificial intelligence methods for brain tumor segmentation and classification using MRI images. They summarized over 100 research papers from 2015 to 2022, focusing on supervised, unsupervised, and deep learning techniques. The review particularly highlights deep learning methods, including CNNs, LSTMs, RNNs, and GANs. The authors emphasize their application in brain tumor segmentation, noting that CNNs are the most widely used. They discuss several datasets, including the BraTS dataset [38,39,40], and outline MRI modalities such as T1 (for anatomical details), T1c (contrast-enhanced for detecting tumors), T2 (for edema and inflammation), and FLAIR (which suppresses fluid signals for white matter lesions). The review identifies gaps in the literature, such as handling noisy data, addressing class imbalances, and the need for more efficient models. It also highlights challenges like overfitting, high computational costs, and generalization issues. A limitation of the review is that it does not address extraction and classification as thoroughly as segmentation. The authors in [36], provide a thorough summary of deep learning applications in brain tumor analysis, covering key techniques such as segmentation and classification. They propose a taxonomy of state-of-the-art methods while addressing existing challenges, limitations, and future research directions. Various models, including CNNs and Fully Convolutional Networks (FCNs), are discussed, with a focus on 2D and 3D CNNs for tasks like tumor segmentation and feature extraction. Public datasets like BraTS [38,39,40], TCGA-GBM [44], and TCGA-LGG [45], and proprietary datasets are highlighted for their critical role in advancing research. The paper also identifies gaps, such as handling variability in tumor features, improving model generalization across institutions, and addressing limitations in dataset quality. It emphasizes challenges like imbalanced datasets, model interpretability, and the need for robust algorithms that perform well in real-world clinical settings. However, a limitation of the review is its limited focus on feature extraction compared to segmentation and classification, which are explored in greater depth.

Table 2 provides a concise overview of the previously analyzed review papers. Each entry outlines the contributions, datasets discussed, primary advantages, limitations, and shared themes among the reviews. This comparative summary helps identify the evolution of deep learning applications in medical imaging, emphasizing common challenges and highlighting the role of datasets like BraTS, which serve as a benchmark for evaluating model performance in tumor-related tasks.

This work, as outlined in the contributions (Section 1.2), seeks to advance understanding of brain tumor detection by thoroughly examining a wide range of studies that encompass all aspects of the field. It includes a global analysis of research trends, a critical evaluation of previous review works, and an in-depth exploration of feature extraction, segmentation, and classification techniques. Additionally, the work offers discussions and insights into future opportunities for innovation. By addressing such aspects, this work highlights a comprehensive, multilevel framework designed to enhance diagnostic accuracy and give a clearer picture on recent achievements. It aims to drive advancements in medical imaging and brain tumor diagnostics, ultimately contributing to improved patient outcomes and clinical decision-making.

## 3. Trends in Image-Based Feature Extraction

This section provides a general overview of feature extraction using deep learning for brain images and analyzes related works in the context of brain tumor detection through deep learning, with a focus on feature extraction techniques. It presents an example process to familiarize readers with the methodology and workflow of feature extraction. Furthermore, it highlights key approaches from related works, such as Autoencoders, GANs, and CNNs, as well as paradigms like transfer learning. For each approach, the discussion addresses the application of feature extraction, the datasets used, and optimization tools or strategies employed to enhance model performance. Consequently, this section aims to offer insights into the role of feature extraction methodologies in advancing effective brain tumor detection within medical imaging.

### 3.1. Overview of Feature Extraction with Deep Learning

Feature extraction from MRI brain images refers to the process of identifying and extracting meaningful patterns or characteristics from raw image data to facilitate further analysis. This step is crucial in transforming high-dimensional image data into a set of informative features that can be used for tasks like segmentation, classification, and detection of abnormalities, such as brain tumors [46,47,48]. The process begins with acquiring MRI images from various datasets, followed by preprocessing steps such as normalization, resizing, and data augmentation to improve data quality. Next, feature extraction techniques, either supervised or unsupervised, such as CNNs and autoencoders, respectively, are applied to derive meaningful features. These features are then used in model training, enabling segmentation or classification of tumors. Finally, model evaluation assesses performance and accuracy on both seen and unseen data, supporting enhanced diagnostic capabilities in clinical settings [49,50,51,52].

To gain a comprehensive understanding of the extraction process, we utilize a Kaggle dataset, referenced as [53], as a primary illustrative example throughout this review. This dataset comprises MRI images organized into four primary categories: healthy brain, glioma tumor, meningioma tumor, and pituitary tumor, with representative examples illustrated in Figure 5. Files for each category are stored in separate folders to facilitate easier labeling and quick access. Upon examining the MRI images of different tumor types, several distinguishing characteristics are observed. Glioma tumors (Figure 5b), for instance, typically present as irregularly shaped, infiltrative masses that often invade surrounding tissues, making their boundaries harder to define. Meningioma tumors (Figure 5c), on the other hand, generally appear as well-defined, convex lesions often attached to the dura mater, which is the outer covering of the brain. Pituitary tumors (Figure 5d), in contrast, tend to appear in the region of the pituitary gland, usually displaying a more localized growth pattern. These variations in tumor appearance are critical in the feature extraction process, as they affect how the features are derived and influence the subsequent classification or segmentation tasks.

Accordingly, each image in these folders is processed to extract features that can be utilized for further analysis. As part of the analysis, we conducted an experiment using an Autoencoder trained on the dataset images. The images were resized to a consistent dimension of 16×16 pixels to ensure uniformity before training the Autoencoder. This standardization is crucial for effective feature extraction and model performance. The training was carried out using a specific set of parameters, which included a hidden layer size of 64 neurons, allowing the model to capture patterns within the data. The Autoencoder was trained for 1000 epochs, with a performance goal set at $1\times 10^{-6}$ to ensure the accuracy of feature representation. After training, we proceeded to map the processed images from each folder. Figure 6 illustrates the results of feature extraction from MRI brain images using an autoencoder, focusing on both healthy brain tissue and three types of brain tumors: glioma, meningioma, and pituitary tumors. The feature patterns are represented through varying hues, with each color line corresponding to a distinct feature set extracted from the images. The hue variations are key in visualizing how the model differentiates between healthy brain tissue and various tumor types, highlighting the differences in the patterns formed by the features.

Upon visual inspection, the feature patterns of healthy brain tissue (Figure 6a) and the three tumor types, glioma (Figure 6b), meningioma (Figure 6c), and pituitary tumor (Figure 6d), can be observed to show both distinct and overlapping characteristics. Some of the tumor types, like glioma, tend to display more irregular and diffused patterns, while meningiomas tend to show well-defined, more localized patterns. Pituitary tumors often have a more focused growth pattern, which is reflected in the feature mapping. These distinct feature patterns, captured by the autoencoder, help the model identify and differentiate between various tumor types.

However, while some clear distinctions between tumor types are visible, there are also notable similarities, particularly between the healthy brain and certain tumor types. These similarities complicate the classification process, as the model may struggle to fully distinguish between them, especially in cases where tumor types exhibit subtle variations in their feature distributions. The overlap in feature patterns underscores the challenges faced in the classification task, as tumors can sometimes share similar characteristics in their appearance, even though they are biologically different.

This observation highlights the strengths and limitations of using the autoencoder for tumor classification. While the autoencoder captures the essential features that distinguish healthy tissue from tumors, the similarity between tumor types, as well as the subtle differences between them, suggests that further refinement of the model is needed. Additional strategies, such as fine-tuning the feature extraction process or incorporating more sophisticated models, could help improve the differentiation between tumor types, especially when the tumors exhibit closely related patterns.

Additionally, we conducted t-distributed Stochastic Neighbor Embedding (t-SNE) analysis to further investigate the relationships between the classes [54]. The results are presented in Figure 7, where scatter plots visualize the outputs of both the original data and the Autoencoder features after PCA-based dimensionality reduction.

In Figure 7a, the scatter plot represents the t-SNE results from the original data, while Figure 7b shows the scatter plot for the Autoencoder features. In these plots, distinct clusters emerge for different classes, suggesting that the Autoencoder has encoded the features while preserving class information. However, overlapping patterns are also observed, indicating that further stages of analysis, such as segmentation or supervised learning, may be necessary to improve classification accuracy and refine feature extraction. This overlap suggests that the Autoencoder may still require fine-tuning based on supervised learning. Incorporating labeled data can enhance the model by improving the separability of class patterns, ensuring more distinct feature representation and classification accuracy.

Overall, the examples of the Autoencoder across different stages of mapping and scatter plot visualizations have provided valuable insights into the distinguishing features of various tumor types. These insights were achieved through rigorous feature extraction and advanced visualization techniques. The results from both the heatmap and t-SNE visualizations highlight the strengths and limitations of the current approach, showcasing its ability to encode some features at a lower level of differentiation between tumor types while also revealing gaps in accuracy. These findings underscore the need for further refinement of the model, emphasizing the importance of improving classification performance for brain tumors based on MRI images in future research. Additionally, these gaps underline and explain some key challenges in feature extraction, which can be outlined as follows [55,56,57,58].

High dimensionality in MRI images often results in overfitting and computational inefficiency, requiring dimensionality reduction techniques to manage effectively without losing critical information.Noise and artifacts in medical images, stemming from variations in acquisition protocols or patient movement, can compromise the quality of extracted features and the overall reliability of the model.Class imbalance in datasets, where certain tumor types are underrepresented, limits the model’s ability to learn features effectively for minority classes, leading to biased predictions.Overlapping feature spaces between tumor classes reduce the model’s ability to distinctly separate classes, hindering accurate classification and visual differentiation.Limited dataset size, a common issue in medical imaging, restricts the model’s capacity to generalize across different populations or imaging conditions, necessitating data augmentation or synthetic data generation.Dynamic variability in tumors, including differences in size, shape, and intensity, poses significant challenges for feature extraction methods to adapt and capture these variations comprehensively.Fine-tuning Autoencoders or any unsupervised learning feature extractors with supervised learning introduces optimization complexity, requiring careful selection of hyperparameters and loss functions to avoid underperformance or convergence issues.

### 3.2. Related Works to Feature Extraction

Among the collected papers, relevant studies have been reviewed in this subsection to analyze how feature extraction methods have been implemented within this specific area of the field. This analysis of related works has been conducted to provide insights into the advancements and approaches utilized in feature extraction. For instance, in [46] the authors utilized radiomic feature extraction for brain tumor classification on multi-modality MRI (T1, T2, FLAIR) scans. The dataset, sourced from Kaggle, comprised 3265 images across four categories: glioma, meningioma, pituitary tumors, and healthy brains. They tested various machine learning models, including Support Vector Machine (SVM), Decision Trees (DTs), XGBoost, and FCN. XGBoost achieved the highest accuracy (88.51%), with the FCN closely following at 87.09%. Optimization tools like XGBoost’s tree-pruning and dropout layers in FCN were employed to enhance performance and prevent overfitting. In ref. [59], the authors present a method for brain tumor segmentation and classification using pre-trained CNNs, specifically AlexNet and GoogleNet. These CNNs are employed in a transfer learning framework where the models, initially trained on large datasets for generic image recognition, are fine-tuned with MRI and CT images to extract features relevant to brain tumor detection. The feature extraction process involves segmenting the MRI and CT images using a combination of morphological operations and CNN feature mapping. Regarding datasets, this work involves BraTS [38,39,40] datasets from and the ISLES [42] dataset, which provide multimodal MRIs and CT images. The extracted features from each CNN model undergo score fusion at the softmax layer, where scores from both AlexNet and GoogleNet are combined. This fused score vector is then used in multiple classifiers, including SVM, DT, and Linear Discriminant Analysis (LDA), to improve classification accuracy. In ref. [60], the study employs CNNs and transfer learning for brain tumor classification using MRI images. Specifically, they utilize the pre-trained VGG19 model, a CNN trained on the ImageNet dataset, to address the small dataset problem inherent in medical imaging. This model is applied to the Contrast Enhanced Magnetic Resonance Imaging (CE-MRI) dataset [61], which includes three types of brain tumors: glioma, meningioma, and pituitary tumors. The authors adopt a block-wise fine-tuning strategy on VGG19 model. This method involves fine-tuning of pretrained models layers progressively and minimizing overfitting during such domain adaptation. For feature extraction, the model learns low-level features (e.g., edges and shapes) in the early layers and progressively captures higher-level, domain-specific features in the deeper layers. In ref. [62], the authors use CNNs combined with Gray-Level Co-occurrence Matrix (GLCM) and Cellular Automata (CA) to detect and segment brain tumors in MRI images. GLCM extracts textural features like Angular Second Momentum and Entropy (ASME), while CA refines texture segmentation. The MRI dataset is sourced from hospitals, with pre-processing steps including intensity standardization and k-means clustering to prepare the images for CNN classification. No specific transfer learning or optimization tools are mentioned, as the focus is on CNN architecture and texture integration for accurate tumor detection. In ref. [63], the authors employ image-based feature extraction for brain tumor segmentation and classification using the T1-weighted contrast-enhanced MRI (T1-WCEMRI) dataset. They segment images with Adaptively Regularized Kernel-Based Fuzzy C-Means (ARKFCM) and perform hybrid feature extraction using GLCM, Local Binary Pattern (LBP), and Histogram of Oriented Gradients (HOG) to capture texture and gradient information. For feature selection, a Genetic Algorithm (GA) combined with K-Nearest Neighbor (KNN) identifies the most relevant features, optimizing classification. These features are then fed into an Autoencoder linked to a custom Deep Neural Network (DNN) for classifying tumor types, including, meningioma, glioma, and pituitary. This GA-based feature selection serves as an optimization tool, enabling the method to achieve a classification accuracy of 97.33%, outperforming traditional CNN and SVM approaches. In ref. [64], the authors applied a CNN combined with semantic segmentation to detect and classify brain tumors in MRI and CT images. Using a T1-WCEMRI images from 233 patients, they segmented tumors and classified them into meningioma, glioma, and pituitary types using a transfer learning method, specifically, employing GoogLeNet. For optimization, they employed Stochastic Gradient Descent Method (SGDM) and specific training settings to enhance accuracy. This approach achieved high accuracy, exceeding 99% for each tumor type, by overlaying CNN and semantic segmentation results to improve classification precision. In ref. [65], the authors used CNNs for feature extraction to detect small brain metastases in T1-WCE 3D MRI images. They developed a custom CNN architecture named CropNet specifically designed to identify metastatic regions from candidate ROI, which were pre-selected using a Laplacian of Gaussian (LoG) approach. The authors used a dataset of 217 T1-WMRI scans from 158 patients with a total of 932 metastases. They employed data augmentation techniques, including random gamma correction and elastic deformation, to increase training sample diversity and enhance model robustness. For optimization, they utilized the Adam optimization algorithm and applied five-fold cross-validation to validate their results. The system achieved high sensitivity, comparable to other state-of-the-art detection frameworks, especially for small metastases. In ref. [66], the authors used a custom CNN and transfer learning with pre-trained models (VGG-16, ResNet-50, and Inception-v3) to classify brain tumors in MRI images. Feature extraction in this work leverages a CNN-based architecture integrated with additional transfer learning within a supervised learning framework. They applied data augmentation on a small Kaggle dataset of 253 MRI images, split into training, validation, and testing sets. The custom CNN, designed to detect various image patterns through convolutional layers and max-pooling, achieved 100% training accuracy. For optimization, they used the Adam optimizer, improving convergence. Transfer learning models were fine-tuned, though the custom CNN showed superior accuracy with less computational demand. In ref. [67], the authors apply feature extraction through a hybrid Autoencoder combined with Bayesian Fuzzy Clustering (BFC) for brain tumor classification using MRI data from the BraTS dataset. They employ the BFC method for segmenting tumor regions, isolating areas of interest like the edema and core tumor. For feature extraction, the authors utilize methods including Wavelet Packet Tsallis Entropy (WPTE), Scattering Transform (ST), and information-theoretic measures, focusing on robust feature representation from the segmented regions. These features are then fed into the DAE model, where the Jaya Optimization Algorithm (JOA) is used for fine-tuning weights to reach an optimal solution in classification tasks. Additionally, the softmax regression is applied as the final layer in the Autoencoder for classifying tumor presence. In ref. [68], the authors utilize a CNN as part of a transfer learning paradigm to identify and classify brain tumors in MRI scans. They employed ResNet-50 and fine-tune it on new data to distinguish between three categories: healthy brain tissue, vestibular schwannoma, and glioblastoma. The dataset includes MRI scans from IXI dataset [41] for healthy images. Images of vestibular schwannoma provided are by the European Cyber-Knife Center (ECKC), and glioblastoma images are from TCIA. To enhance the interpretability, the authors integrated Gradient-weighted Class Activation Mapping (Grad-CAM), a tool for visual highlights the regions of interest in images witch CNN focused on during classification. Additionally, a Bayesian neural network approach was implemented to gauge model confidence in its predictions. This approach allowed the model to produce predictions only when confident, increasing accuracy while reducing uncertainty. In ref. [69], the authors use a Bi-directional Modified GLCM (Bi-MGLCM) to extract texture features from MRI scans, analyzing symmetry between brain hemispheres of normal and abnormal health status. The study employs two datasets: a clinical dataset of 214 MRI images from Al-Imamain Al-Kadhimain Medical City in Iraq and BraTS dataset [38,39,40]. LSTM optimized with Adam algorithm is used for the classification process. It achieves high accuracy of 96.3% on the collected dataset and 98.9% on BraTS outperforming other classifiers. No advanced techniques like transfer learning or Autoencoders are used. In ref. [70], the authors develop a hybrid model combining CNN and Deep Watershed Autoencoders (DWA) for brain tumor detection and classification from MRI images. The model leverages CNNs for feature extraction, focusing on spatial relationships within images, and applies DWA for segmentation and classification. Using a dataset of 3650 MRI images from BraTS [38,39,40] and ISLES [42], the model achieved a high accuracy of 98% and a Dice Similarity Coefficient (DSC) of 96.55%. Optimization included data augmentation (rotation, shifting, and magnification), and training was conducted with TensorFlow and Keras, enhancing robustness and effectiveness for early tumor detection. In ref. [71], the authors present a brain tumor MRI image classification method called Convolutional Dictionary Learning with Local Constraint (CDLLC), using transfer learning based on AlexNet within a CNN framework for feature extraction. CDLLC enhances discriminative feature representation by applying multi-layer dictionary learning and sparse coding within CNNs. The study utilizes two datasets, the Cheng dataset and the Repository of Molecular Brain Neoplasia Data (REMBRANDT) dataset [72], for classifying different brain tumor types. An iterative optimization scheme, including SGDM and Graph Laplacian Regularization (GLR), are applied to maintain local structure and improve classification accuracy. In ref. [73], the authors design a computer-aided diagnostic network for classifying MRI brain tumor images using a Gabor-modulated CNN (GMCNN). The network integrates Gabor filters within CNNs to capture spatial and orientation-specific features, enhancing accuracy with fewer parameters. Using the BraTS [38,39,40], containing high-grade and low-grade Glioma MRIs, they also leverage transfer learning by fine-tuning models like AlexNet, VGG, and ResNet for comparative analysis. Optimization is performed using the Adam algorithm with L2 regularization, along with Leave-One-Patient-Out (LOPO) cross-validation to ensure robust performance across patients. This GMCNN approach improves both efficiency and classification accuracy. In ref. [74], the authors developed Hahn moment Pulse-Coupled Neural Networks and CNN (Hahn-PCNN-CNN), a framework for multi-modal brain image fusion that enhances clinical diagnostics by effectively combining features from structural and functional brain images. Utilizing CNN-based feature extraction with transfer learning from GoogLeNet and Hahn moments in the PCNN-based fusion module, the framework captures complex textures and metabolic information while minimizing image artifacts. The model was trained on 8000 images from Harvard Medical School, and it outperformed other methods in preserving structural and functional details. Optimizations, including cross-entropy and Multi-Scale Structural Similarity Index Measure (MS-SSIM) loss functions, were employed to reduce computational complexity and refine image quality. In ref. [75], the authors applied a CNN with machine learning classifiers, specifically SVM with Radial Basis Function (SVM-RBF), Random Forest (RF), and Extreme Learning Machine (ELM), for feature extraction, segmentation, and classification of brain tumors in MRI images. Feature extraction was primarily conducted through CNN layers that were trained to identify relevant features in the MRI scans, enhancing tumor segmentation accuracy by utilizing Region Proposal Networks (RPNs) for precise tumor localization. The authors used two datasets: A Kaggle dataset, containing 3174 MRI images with glioma, meningioma, and pituitary tumors, and the Figshare dataset with 3064 T1-WCEMRI. The proposed model achieved high accuracy in classification, recording 98.3% on Kaggle dataset and 98.0% on the Figshare dataset. Furthermore, they employed optimization techniques including the Adam optimizer with fine-tuned hyperparameters, to improve model performance on training and validation metrics. In ref. [76], the authors propose a hybrid deep CNN and transfer learning model, specifically utilizing the ResNet-152 architecture, for enhanced brain tumor detection and classification from MRI images. For feature extraction, they apply GLCM methods to obtain texture features like contrast, energy, correlation, homogeneity, and entropy, which are critical for distinguishing normal and tumor tissues. The dataset employed is the BraTS dataset [38,39,40], comprising both normal and abnormal brain images. The model benefits from the COVID-19 optimization algorithm (CoV-19 OA), which is used to fine-tune the model’s parameters, enhancing accuracy and reducing computational complexity. In ref. [77], the authors applied feature extraction by combining FCM clustering with a deformable CNN model, termed Adaptive Fuzzy Deformable Fusion (AFDF) segmentation. This model was enhanced through optimization using an Adaptive Coefficient Vector-based Deer Hunting Optimization Algorithm (ACV-DHOA), targeting brain tumor classification in MRI images. The ACV-DHOA optimizes both the segmentation (using FCM and snake deformation) and the classification model’s convolutional layers and hidden neurons, aiming to improve segmentation and classification accuracy. The dataset used in this study comprised MRI brain images sourced from Kaggle, including 253 images categorized into abnormal (155) and normal (98) classes. In terms of optimization tools, the ACV-DHOA was a central component, fine-tuning key parameters in the segmentation and classification models. The optimized CNN was then used alongside an ensemble classifier integrating DNN, Autoencoders, and SVM to improve classification accuracy and generalization. In ref. [78], the authors developed a Bayesian Depth-Wise CNN (BDWCNN) to classify brain tumors in MRI images, combining depth-wise separable convolutions with Bayesian modeling for improved accuracy and efficiency. They employed preprocessing steps such as resizing, normalization, and augmentation followed by depth-wise separable convolutions for feature extraction from tumor regions, which reduced computational costs compared to standard CNNs. The model was trained and validated on MRI images from the BraTS and IXI datasets [38,39,40,41], covering four classes: glioma, pituitary tumor, meningioma, and no tumor. Using the Adam optimizer, dropout, and evaluation metrics like accuracy and F1-score, the model outperformed other CNN-based models, including ResNet and MobileNet.

Table 3 provides a comparative overview of various state-of-the-art deep learning techniques applied to brain tumor feature extraction. It summarizes key methodologies, feature types, and performance metrics across benchmark datasets, highlighting advancements and limitations within the field.

Figure 8a–c elucidates the utilization of various algorithms and datasets in the realm of brain tumor feature extraction, derived from Table 3. This analysis provides an approximate overview of the landscape of methodologies employed in this domain. The findings reveal a dominant trend in algorithm usage, with CNNs and transfer learning underscoring their popularity and effectiveness in this field. These methods appear to be the most frequently mentioned, indicating their widespread adoption and proven success in extracting relevant features for brain tumor analysis.

Hybrid CNNs also emerge as a significant approach, suggesting a growing interest in combining various techniques to improve accuracy and adaptability. Custom architectures are also noteworthy, pointing to a rising trend towards developing tailored approaches for specific tasks. While Autoencoders and GMCNNs are utilized less frequently, their mention highlights their niche application, indicating that these methods may be leveraged in specific contexts or for particular feature extraction challenges. Additionally, LSTM networks, although mentioned less often, are noted for their unique capabilities in handling sequential or temporal data, underscoring their specialized role in the domain.

In terms of datasets, the BraTS dataset emerges as the most commonly used, reflecting its prominence and critical role in brain tumor research. This aligns with its widespread use in the medical imaging community. T1-WCEMRI and ISLES follow, reinforcing their significance in the study of brain tumors, with each offering unique data characteristics beneficial for feature extraction. Other datasets, such as the Kaggle MRI, Cheng, REMBRANDT, IXI, ECKC, TCIA, and the Harvard Medical School dataset, show limited use, suggesting potential areas for expansion in future studies. This limited use may point to specific challenges or limitations related to these datasets, such as size, accessibility, or variability in quality. However, these datasets remain valuable resources for advancing research.

Moreover, personal clinical data represents a comprehensive data source that may enhance the robustness and applicability of feature extraction methods. Leveraging clinical data could bridge the gap between research and real-world applications, providing more personalized and accurate insights into brain tumor diagnostics. In conclusion, Figure 8a–c presents a clear snapshot of the prevalent methods and datasets in brain tumor feature extraction, highlighting both established practices and emerging trends. While some methods and datasets dominate the field, there are significant opportunities for future exploration and innovation, particularly in expanding the use of alternative datasets and advanced techniques.

## 4. Trends in Image Segmentations

This section provides a comprehensive overview of tumor detection in brain images using deep learning, with a particular focus on segmentation techniques. The initial part introduces essential concepts and widely adopted methods for brain image segmentation, establishing a global perspective on the field. The second part delves into related works, examining various approaches and methodologies that have been pursued to enhance the accuracy and efficiency of MRI brain image segmentation.

### 4.1. Overview of Image Segmentation

Segmentation of MRI images for brain tumor detection involves partitioning the image into meaningful regions to isolate and identify tumors within the brain [79,80,81,82,83,84,85]. This process is critical for accurate diagnosis, treatment planning, and monitoring the progression of the disease. Deep learning has significantly advanced this field by automating the segmentation process and achieving higher precision than traditional methods. By leveraging neural network architectures, such CNNs and transfer learning like U-Net, deep learning models can understand patterns allowing for the precise delineation of tumor boundaries and detailed analysis of tumor structures. One of the key advantages of deep learning-based segmentation is its high level of accuracy and precision, which improves diagnostic outcomes by clearly identifying tumor boundaries. Deep learning models also offer capabilities for 3D segmentation, which provides volumetric information crucial for understanding tumor morphology. This is vital for treatment planning, including surgical strategies and radiation therapy, as it gives a comprehensive view of the tumor’s size and location within the brain. The advancements in deep learning have thus made segmentation of MRI images a cornerstone in modern brain tumor detection and treatment, improving both reliability and clinical outcomes.

In this work, a segmentation example has been prepared using the same Kaggle dataset [53] previously utilized for feature extraction illustrations (see Section 3.1). This example employs a modified ResNet-18 architecture, repurposed through transfer learning to perform segmentation on the dataset. The process involves adjusting ResNet-18 to suit the segmentation task for the specified number of classes.

Figure 9 explores the network’s internal representations (i.e., activations) from ResNet layers 5, 10, and 20, respectively. The extracted activations in Figure 9b–d illustrate how the network progressively segments and highlights relevant regions, particularly focusing on specific areas of interest (e.g., brain health states) with color-coded hues. The original image in Figure 9a is presented alongside these activations to demonstrate the model’s ability to concentrate on important tumor patterns. Notably, the color intensities of the tumors are preserved to some extent, maintaining both their location and gradual tonal variations. Furthermore, the skull shape becomes increasingly faded in deeper layers, allowing the network to focus on patterns essential for classification. These layers capture distinctive tumor characteristics, such as localized regions with a sometimes-green circle and sometimes orange circle surrounded degraded yellow, blue or green hues, which are crucial for distinguishing between different brain health states.

This visualization highlights the network’s ability to abstract meaningful features while reducing irrelevant details, such as the skull shape, thereby enhancing the segmentation task. Additionally, an interesting animation example has been created to visualize the full hidden layers of ResNet (layers 1 to 69) during a brain tumor classification task. The visualization begins with the original input image (layer 1) and progresses through each layer (layers 2 to 69) of the network, highlighting how ResNet transitions from detecting low-level features, such as edges and textures, to extracting high-level abstract patterns critical for identifying tumor-specific regions. In the early layers (e.g., layers 2–10), the network retains much of the spatial structure of the input image, focusing on general details like edges, contours, and textures, functioning as foundational feature extractors. As the visualization progresses to intermediate layers (e.g., layers 11–40), the activations emphasize localized patterns and begin filtering out irrelevant components such as the skull while concentrating on regions associated with tumor features and other diagnostic areas. Finally, in the deeper layers (e.g., layers 41–69), the network hones in on highly abstract and classification-relevant features, discarding most of the background and non-essential details to focus entirely on subtle and critical tumor-specific patterns.

These deeper layers are crucial for enabling the model to generalize its understanding and achieve accurate classification results. This visualization offers a comprehensive and intuitive perspective on how CNNs process medical images, showcasing the progressive refinement of features as the data flow through the network, making it a valuable tool for understanding decision-making in medical image analysis. For further details, refer to [86].

While this example represents a foundational approach to image segmentation, the process could greatly benefit from further enhancements. To achieve optimal performance in image segmentation, it is crucial to consider a variety of techniques that enhance both the input data quality and the model’s ability to generalize effectively. These methods address common challenges such as noise, variability in image quality, and overfitting, which can significantly impact segmentation accuracy. By incorporating advanced pre-processing steps and strategic augmentation techniques, the segmentation process can be refined to produce more reliable and precise results. Below is a comprehensive list of potential improvements that can be applied to elevate the overall performance of image segmentation models [87,88,89,90].

Pre-processing techniques like skull removal, noise reduction, contrast enhancement, normalization, resizing, and artifact removal improve image quality and provide cleaner inputs for segmentation models.Image augmentation methods, including rotation, scaling, flipping, elastic deformation, and color jittering, diversify the training data, enhancing the model’s ability to generalize and reducing overfitting.Using diverse and multi-modal data, such as combining MRI and CT scans or incorporating datasets from varied demographics, improves the model’s adaptability to different scenarios.Advanced segmentation methods, like region-growing algorithms, multi-scale analysis, and domain-specific neural networks, help achieve more accurate and context-aware segmentations.Post-processing techniques, such as removing false positives, smoothing boundaries, and applying morphological operations, refine segmentation outputs for better accuracy.Regular evaluation of metrics like the dice coefficient, Intersection over Union (IoU), and precision-recall ensures ongoing optimization, while ensemble methods combine outputs from multiple models for improved robustness and reliability.

### 4.2. Related Works to Image Segmentation

Among the collected papers, an additional set of papers completely different from those in Section 3.2 was selected for the analysis of trends and gaps in MRI brain image segmentation, following the methodology described in Section 1.1. The goal is to identify emerging patterns, highlight underexplored areas, and provide insights that can guide future research in this field. For instance, in [91], the authors present a brain tumor classification framework that combines Variational Autoencoders (VAEs) and GANs to address limited MRI data. By generating high-quality synthetic MRI images, this approach enhances the training dataset and improves classifier accuracy. Using VAEs and GANs for data generation, the model leverages domain-specific knowledge without explicitly applying segmentation. Informative noise vectors from the VAE’s latent space feed into the GAN, producing realistic MRI images and avoiding issues like mode collapse. The study uses a Figshare brain tumor MRI dataset, containing 3064 images of glioma, meningioma, and pituitary tumors. Optimization is performed with the Adam optimizer and hyperparameter tuning on a transfer learning network ResNet50, achieving a classification accuracy of 96.25% and demonstrating the framework’s potential for medical imaging. In ref. [92], the authors present a deep learning framework using CNNs to detect and classify cancerous brain tumors from MRI images. The model architecture includes convolutional, ReLU, pooling, and fully connected layers, allowing for effective feature extraction and tumor region identification. Preprocessing techniques like noise reduction, skull stripping, and morphological operations enhance image quality for better segmentation. The study employs an MRI dataset with multiple tumor types, optimized with the Adam optimizer and cross-validation for improved performance, and achieves high accuracy, precision, and recall. In ref. [93], In this study, the authors present a deep learning approach using a CNN model to classify brain tumors in MRI images. They integrate transfer learning models AlexNet and VGGNet, two widely recognized CNN architectures, to leverage their distinct feature extraction strengths. The model’s architecture involves running MRI images through each network branch in parallel, with extracted features combined at a final stage to improve classification. Key to this approach is the use of the Adam optimizer, which refines learning rates throughout training, enhancing the model’s convergence and efficiency. The datasets used include the Figshare brain tumor dataset, with over 3000 images across tumor types, and the TCIA dataset, featuring thousands of glioblastoma images. The model’s performance, validated with k-fold cross-validation, achieves high classification accuracies of 99.14% for binary and 98.78% for multi-class tumor types, showing substantial improvement over traditional methods and demonstrating its potential as a reliable diagnostic support tool for radiologists. In ref. [94], the authors propose a brain tumor segmentation approach that combines both learned and nautical extracted features based human intervention to achieve more accurate segmentation in MRI images. They employ SegNet, a pretrained transfer learning network CNN, to perform semantic segmentation of the Region Of Interest (ROI) in MRI images. The learned features are extracted from SegNet’s deconvolution layer, while analytical texture features are derived from GLCM. The approach only segments tumor-relevant areas (edema, necrosis, and enhanced tumor) to reduce computational complexity. The authors applied their method to the BraTS dataset, which consists of multi-modal MRI scans of brain tumor patients. Each image was segmented into distinct tumor regions, and the model was evaluated based on segmentation accuracy in these regions. For optimization, the authors adjusted the parameters of the SegNet network to achieve the best performance in segmenting ROIs, and they optimized the DT classifier used to classify each pixel within the segmented areas. The combination of SegNet and GLCM features yielded an enhanced F-score, demonstrating improved segmentation accuracy compared to ordinary SegNet. In ref. [95], the authors applied image segmentation for brain tumor detection using hybrid deep learning techniques. They combined CNNs with transfer learning U-Net architecture, enhanced by Inception modules that capture tumor features at multiple scales. They also used depth-wise separable convolutions to improve efficiency and segmentation accuracy. The segmentation model was trained on the Medical Image Computing and Computer-Assisted Intervention (MICCAI) conference challenge BraTS dataset, comprising MRI scans from 335 patients across four modalities (T1, T2, T1-Gd, and FLAIR). Preprocessing included standardizing image dimensions and skull-stripping to isolate brain tissue. Optimization tools included the Adam optimizer, batch normalization, and data augmentation (flipping, zooming, shearing). Their depth-wise separable hybrid model outperformed baseline U-Net, achieving higher dice coefficient, sensitivity, and specificity. In ref. [96], the authors applied transfer learning with CNNs, specifically, VGG-16, ResNet-50, and Inception-v3 models, for brain tumor classification in MRI images. Using a dataset of 253 MRI images from Kaggle, they employed preprocessing steps like resizing, segmentation, and data augmentation to address dataset limitations and improve model performance. Among the models, VGG-16 achieved the highest accuracy (96%) with minimal loss. Optimization techniques such as tuning learning rates and using data augmentation further enhanced accuracy and generalization. In ref. [97], the authors proposed a method for early brain tumor detection using MRI images. They applied Adaptive Histogram Contrast Normalization with Learning-based Neural Quantization (AHCN-LNQ) for image preprocessing to enhance contrast. For segmentation, they used Otsu thresholding to distinguish tumor regions by optimizing pixel separation. Using MRI brain images, they applied LNQ for classification, achieving high accuracy, specificity, and precision, outperforming methods like k-means and standard neural networks. In ref. [98], the authors present a hybrid CNN-LSTM networks to improve brain tumor detection in MRI images. The CNN extracts spatial features of tumors, while LSTM layers capture temporal dependencies, enhancing classification accuracy. Image segmentation is incorporated into preprocessing to focus on tumor regions by using cropping and resizing techniques. Extreme point calculation and bicubic interpolation ensure consistent image dimensions, preparing them for feature extraction. The model’s dataset, obtained from Kaggle, includes 253 MRI images (98 non-tumor and 155 tumor), providing a balanced foundation for training and evaluation. Adam optimization, along with cross-entropy loss, supports efficient model training across 100 epochs, achieving 99.1% accuracy, 98.8% precision, 98.9% recall, and an F1-score of 99.0%. In ref. [98], the authors propose a Graph Attention Autoencoder-CNN (GATE-CNN) model for classifying brain tumors from MRI images. By combining a GATE with a CNN, the model captures both structured and unstructured data, enhancing classification accuracy. The GATE module transforms MRI images into graphs by calculating attention values between neighboring pixels, which is then reconstructed and passed to CNN for final classification. The model is tested on three Kaggle MRI datasets: benign vs. malignant tumors, glioma vs. pituitary tumors, and normal vs. abnormal brain images, demonstrating robust classification performance across varied tumor types. To optimize training, the Adamax optimizer is applied alongside dropout layers in the CNN to prevent overfitting. This dual approach, leveraging graph structures with CNN processing, achieves high accuracy and efficiency, improving MRI-based brain tumor classification. In ref. [99] the authors propose a multi-modal approach for brain tumor detection using a custom 17-layer CNN for segmentation, combined with a modified transfer learning pretrained MobileNetV2 for feature extraction via transfer learning. Classification is handled by a multiclass SVM. The method is tested on BraTS 2018 and Figshare MRI datasets, which include T1-weighted images of meningiomas, gliomas, and pituitary tumors. An entropy-based feature selection method is applied to optimize the extracted features, and the Adam optimizer is used to train the CNN, achieving high accuracy and outperformance in tumor detection and classification. In ref. [100], the authors introduce a Chaotic Whale Cat Swarm Optimization (CWCSO)-enabled CNN approach for brain tumor segmentation and classification from MRI images. They apply Fractional Probabilistic Fuzzy Clustering (FPFC) for segmenting tumor regions, followed by feature extraction using techniques like wavelet transforms and the Significant Local Optimal Oriented Pattern (SLOOP). The CNN model utilizes transfer learning to classify tumor types (non-tumor, core tumor, edema, enhanced tumor), and its weights and biases are optimized through CWCSO. This combination improves model accuracy and convergence. Using the BraTS dataset, which contains multi-modal MRI scans, the model achieved high performance: 98.59% specificity, 95.52% accuracy, and 97.37% sensitivity, indicating its effectiveness in brain tumor classification. In ref. [101], the authors develop a deep learning-based model for brain tumor diagnosis and classification using MRI images. The model employs median filtering to reduce noise and uses morphological operations (dilation and erosion) for tumor region segmentation. For feature extraction, it combines handcrafted texture features using GLCM with deep features from transfer learning network VGGNet, a CNN model. Classification is conducted via an ANN, optimized using the Artificial Fish Swarm Optimization (AFSO) algorithm to enhance performance. The model is evaluated on a Figshare dataset with three tumor classes: meningioma, glioma, and pituitary, showing high accuracy in classification. The study while utilizes transfer learning, it focuses on feature fusion and AFSO-based optimization. In ref. [102], the authors present an improved Deep Learning Cascade Regression (DLCR) model for brain tumor segmentation in MRI images. The approach includes preprocessing with a CNN to remove noise and normalize intensity, followed by feature extraction using a Gaussian Mixture Model (GMM) to isolate key characteristics. The DLCR model operates in three stages: random selection of image patches, a distance-based attention mechanism to locate tumor regions, and final segmentation. Using the BraTS dataset, the model demonstrated enhanced segmentation performance, surpassing methods like machine learning, deep learning, and ELM with Local Receptive Fields (ELM-LRF). Optimized with the ADAM optimizer, DLCR achieved higher sensitivity, specificity, precision, recall, PSNR, and lower RMSE, confirming its accuracy and efficiency in brain tumor segmentation. In ref. [103], the authors developed Brain Cancer MRI–VGG16-based Ensemble of Machine Learning Techniques (BCM-VEMT), a system to classify brain cancer types from MRI images using deep learning and an ensemble of machine learning techniques. A pre-trained VGG-16 CNN with transfer learning was used to extract deep features from MRI images, classifying them into four categories: glioma, meningioma, pituitary, and normal healthy brain tissue. The dataset, comprising 3787 MRI images from public sources, was augmented through rotation, shifting, flipping, shearing, and brightness adjustments to enhance performance. To further optimize accuracy, the authors applied a grid search for tuning hyperparameters across multiple classifiers, which were combined in a weighted ensemble based on individual classifier accuracy. This ensemble approach improved classification results, achieving an overall accuracy of 98.42%, with early stopping used to prevent overfitting during CNN training. In ref. [104], the authors used CNNs and image enhancement techniques for brain tumor classification from MRI images. They applied Gaussian blur-based sharpening and Contrast-Limited Adaptive Histogram Equalization (CLAHE) to improve image quality, allowing the CNN to capture tumor-specific features more effectively. The study utilized a dataset of 7023 grayscale MRI images from Figshare, SARTAJ, and BR35H, representing glioma, meningioma, pituitary tumors, and non-tumorous cases. For optimization, the model used the Adam optimizer with ReduceLROnPlateau for dynamic learning rate adjustment, as well as dropout, L1, and L2 regularizations to reduce overfitting. This approach achieved a high accuracy of 97.84%, surpassing other pre-trained models such as VGG16 and ResNet50. Instead of manual segmentation, the CNN extracts feature directly from enhanced images, streamlining tumor detection without specific localization steps. In ref. [105], the authors refined a YOLOv7-based CNN model for classifying brain tumors from MRI images. They used transfer learning, initializing the model with pre-trained COCO dataset weights, to detect gliomas, meningiomas, and pituitary tumors effectively. To enhance feature extraction, they integrated Convolutional Block Attention Module (CBAM), Spatial Pyramid Pooling Fast+ (SPPF+), and Bi-directional Feature Pyramid Network (BiFPN), improving the model’s ability to detect tumors of varying sizes. The authors employed an open-source MRI dataset from Kaggle with 10,288 labeled images and used data augmentation to expand it, boosting generalizability. Optimization techniques, including the Adam optimizer and binary cross-entropy loss, further improved accuracy. In ref. [106], the authors propose a brain tumor diagnosis method using a CNN optimized by an improved version of the political optimizer. This CNN segments MRI images to detect brain tumors, with hyperparameters like kernel size and dropout rate fine-tuned via the optimizer. The Figshare brain tumor dataset, featuring meningioma, pituitary, and glioma tumor images, is used for training and validation (80–20 split). Preprocessing steps, skull removal, noise reduction, and contrast enhancement, prepare the data, and image augmentation strengthens model robustness. The enhanced political optimizer uses opposition-based learning and chaos theory to refine CNN architecture, achieving a high classification accuracy of 96%. These results underscore the CNN’s effectiveness in accurate tumor segmentation. In ref. [107], the authors developed a Feature Enhanced Stacked Autoencoder (FESAE) for detecting brain diseases in MRI images. They used the SAE to improve feature extraction, adding an enhancement function to capture non-trivial features for higher accuracy. The approach combined spatial and frequency domain information using Discrete Wavelet Transform (DWT) and RGB channelization to enhance feature details, leveraging autoencoders for image segmentation in deep learning. The dataset, sourced from Kaggle and Harvard Medical College, included 2000 MRI images across four classes: normal, tumor, brain stroke, and Alzheimer’s. To optimize the model, the authors used a unique enhancement function that refines feature extraction by distinguishing neurons based on activation energy, alongside regularization techniques. The FESAE achieved a classification accuracy of 98.61%. In ref. [108], the authors proposed a Multi-Modal Generative Adversarial Network (MMGAN), a brain tumor segmentation method that combines with GANs with a deep residual U-Net architecture. Replacing conventional CNN layers with deep residual networks, the model enhances feature extraction and stability in training. GANs drive the segmentation by having a discriminator evaluate and refine the generator’s output for greater accuracy. The authors utilize the BraTS dataset, containing multimodal MRI images (T1, T1c, T2, FLAIR) with expert-labeled segmentation. To optimize performance, they incorporate an attention mechanism (SEblock) to prioritize valuable feature channels and use a multiscale L1 loss in the adversarial network to better align the generated segmentations with ground truth. In ref. [109], the authors apply CNNs and transfer learning, specifically fine-tuning EfficientNet models (EfficientNetB0-B4), for classifying brain tumor types in MRI images. Using pre-trained ImageNet weights, they modify EfficientNets to classify glioma, meningioma, and pituitary tumors. Data augmentation, including rotation and flipping, enhances the training set and helps prevent overfitting. The CE-MRI brain tumor dataset from Figshare serves as the data source. Optimized with the Adam optimizer and regularization via dropout layers, the models achieved high accuracy, precision, and recall. Grad-CAM visualization further highlights key tumor regions in MRI images, underscoring the model’s diagnostic capability. In ref. [110], In this work, the authors used a CNN, specifically a modified AlexNet architecture, for noninvasive grading of glioma brain tumors using MRI. The CNN model classifies brain MRI images into three categories: healthy, low-grade glioma, and high-grade glioma, aiming to provide a diagnostic alternative to invasive biopsies. The dataset, sourced from the TCIA, includes FLAIR-weighted MR images of 130 subjects annotated by experts. A total of 4069 2D MRI slices were processed, focusing on lesion regions to improve classification accuracy. Key optimizations included hyperparameter tuning for learning rate, batch size, weight decay, and dropout rate, along with preprocessing steps like contrast normalization and image down-sampling.

Table 4 summarizes these recent advancements in brain tumor segmentation and classification using various deep learning techniques. This table highlights the segmentation methods, underlying paradigms, advantages, limitations, and datasets utilized across studies. By comparing these approaches, Table 4 provides insight into the strengths and potential constraints of each method, offering a clear overview of how different models leverage specific architectures, datasets, and optimization techniques to improve brain tumor diagnosis from MRI images.

Figure 10 offers a detailed overview of the methods, paradigms, and datasets used in the study of brain MRI image segmentation, providing clarity through categorical grouping. The data are derived from an analysis of Table 4.

Figure 10a shows the frequency of various methods employed in the study, with transfer learning techniques, including widely recognized architectures such as AlexNet, VGGNet, and ResNet-50, standing out as the most prominent. This highlights the central role of transfer learning in brain MRI segmentation tasks. CNNs also appear frequently, underscoring the effectiveness of their architecture for specific segmentation applications. Other methods, such as GANs, Autoencoders (including VAE and FESAE), Inception, LSTM, and Graph Attention, are used less often, reflecting their specialized roles in addressing specific challenges within brain MRI image segmentation.

Figure 10b categorizes these methods into broader paradigms, with transfer learning again dominating the chart, emphasizing its extensive use in brain MRI segmentation. This paradigm demonstrates the value of leveraging pre-trained models to enhance segmentation performance.

Finally, Figure 10c illustrates the frequency of datasets used in the study. Figshare emerges as the most frequently utilized dataset, followed by Kaggle and BraTS, with Figshare notably being the most widely used, indicating its broad applicability in brain MRI segmentation. Kaggle and BraTS also play significant roles, with BraTS being a well-known source for brain tumor segmentation. Datasets like TCIA, MICCAI BraTS, and Custom MRI brain images are less frequently mentioned, suggesting their more specialized or niche applications in the field.

## 5. Trends in Image Classification

Moving on to classification tasks, this section provides an overview of classification-related examples, key concepts, and the main challenges. Additionally, a separate subsection will focus on related works, exploring methods and challenges specific to classification. Similar to the previous sections on extraction and segmentation, this section reviews twenty articles in the field of classification different from those in previous sections (Section 3.1 and Section 4.1), offering a comprehensive examination of the methods and challenges addressed in the literature.

### 5.1. Overview of Classification

The classification of brain tumors using deep learning and medical imaging is a cutting-edge technology that utilizes advanced algorithms to analyze and interpret complex patterns within imaging data, primarily from modalities like MRI and CT [111,112,113,114,115,116]. Deep learning models are adept at automatically extracting hierarchical features, enabling accurate differentiation between tumor types, grades, and malignancy levels. These models are trained on extensive labeled datasets, allowing them to surpass traditional methods in accuracy, speed, and reliability. The integration of deep learning in medical imaging facilitates early and precise diagnosis and also supports tailored treatment planning, offering significant advancements in patient care.

The pipeline for classifying brain tumors using deep learning and medical imaging typically involves several key stages. First, the process begins with data acquisition, where high-quality brain imaging scans, such as MRI or CT images, are collected. These images undergo pre-processing steps, such as noise reduction, normalization, extraction, and segmentation, to enhance image quality and focus on regions of interest, such as the tumor area. The pre-processed images are then fed into a deep learning model, often a CNN, which extracts features automatically. This model is trained using a labeled dataset, where each image is paired with a corresponding tumor classification label (e.g., benign, malignant, or specific tumor types like glioma or meningioma). Once trained, the model evaluates unseen images to classify the tumor with high precision. Post-processing steps may include visualization of predictions and uncertainty analysis to ensure clinical interpretability and reliability. The automated pipeline of deep learning-based image classification streamlines the diagnostic process, reducing manual effort and improving diagnostic accuracy.

The example in Figure 11 is generated from the same previously trained ResNet in Section 4.1 model used in prior feature segmentation examples. Here, the original validation data are projected onto a lower-dimensional space using PCA, while the ResNet layer activations are also collected and undergo similar dimensionality reduction steps. The intent is to visualize how deep learning, particularly through transfer learning, enhances classification performance by transforming feature space. In Figure 11a, the data scatter plot represents features directly derived from the original validation images. This plot shows that the samples appear less agglomerated and less distinguishable, which can be attributed to the raw features lacking the discriminative power necessary for robust classification. By contrast, Figure 11b displays the scatter plot of ResNet activation mappings. Here, the samples are more clustered and well separated, reflecting the impact of the learned feature representations within the ResNet model. This difference highlights how transfer learning enriches feature representation, making the clusters of classes more distinguishable and improving classification performance.

Despite using transfer learning in Figure 11b, some degree of overlap remains, indicating that the model requires further refinement and optimization. Nevertheless, compared to the results achieved with the Autoencoder (Section 3.1), the ResNet’s mapping demonstrates improved accuracy and better performance. The observed overlap and suboptimal clustering can be attributed to several challenges inherent in current deep learning models and feature extraction techniques, as outlined in [117,118,119,120,121,122,123] and summarized in the following points.

While transfer learning enhances feature representation, the learned features may still lack the task-specific optimization needed to fully distinguish between classes, leaving overlapping in certain areas.Despite improvements, activation mappings can still include noise or redundant features, which obscure meaningful patterns and contribute to overlapping clusters.Classes with inherent similarities or subtle differences often remain overlapping, as the model may struggle to fully capture or highlight these distinctions in the feature space.Pre-trained models like ResNet are often optimized for general datasets, and without sufficient fine-tuning, their learned features may not align perfectly with the target domain, affecting class separability.The clustering performance is influenced by pre-processing steps and the dataset’s size and diversity; insufficient pre-processing or limited data can result in incomplete feature representation.Even with transfer learning, certain features may remain overlapping due to shared properties among classes, requiring further refinement, such as advanced loss functions or additional task-specific layers.

### 5.2. Related Works to Image Classification

An interesting body of work focusing on brain tumor detection and classification has been conducted in the literature. For instance, in [111], the authors fine-tuned the Segment Anything Model (SAM) using adapter modules for glioma segmentation in brain MRI images, employing transfer learning. A private dataset of 489 multimodality MRI images from the First Affiliated Hospital of Zhengzhou University was used. They froze SAM’s backbone network, training only adapters and the mask decoder with Adam optimization. The method achieved a Dice score of 87.33%, outperforming state-of-the-art models, demonstrating improved segmentation accuracy with efficient computational requirements. In ref. [112], the authors applied CNNs for brain tumor classification using a Kaggle dataset of 3060 MRI images. The images were preprocessed by resizing and converting to grayscale before being split 80:20 for training and validation. A 5-layer CNN architecture included convolutional layers with Leaky ReLU, max pooling, dropout layers to prevent overfitting, and a sigmoid output for binary classification. The model achieved a 97% validation accuracy, 97.7% precision, 96.5% recall, and a 97.1% F1-score. Optimization included dropout layers and data splitting. In ref. [113], the authors utilized CNNs and transfer learning, to detect brain tumors from MRI images. Pre-trained models like EfficientNetB1, VGG16, ResNet50, MobileNetV2, Inception, and ViT were applied to a dataset of 3200 images categorized into glioma, meningioma, pituitary tumor, and no tumor. Data augmentation and preprocessing improved model generalization. An ensemble model combining EfficientNetB1 and SCNN achieved a classification accuracy of 98%, outperforming individual models. Optimization tools such as the Adam optimizer, early stopping, and learning rate adjustments further enhanced performance. In ref. [114], the authors utilize FCNs with U-Net architecture for brain tumor segmentation in MRI images, leveraging deep learning for semantic segmentation. Using the BraTS dataset with multi-contrast MRI images (T1, T2, and FLAIR), the approach integrates preprocessing steps like zero-mean normalization and data augmentation for consistency. A hierarchical model design applies a 9-layer U-Net for initial segmentation and two 7-layer U-Nets for refining tumor details, followed by VGG16-inspired post-processing to enhance accuracy. Performance is evaluated using the Dice coefficient, achieving high segmentation precision, and results are visualized with Matplotlib and Seaborn, showcasing robust clinical applicability. In ref. [115], the authors evaluated lightweight Vision Transformers (ViTs) and Deep CNNs for brain tumor detection in MRI scans, comparing ViT, MobileViT, Swin Transformer, and ConvNeXt. They used a curated MRI dataset (e.g., SARTAJ, Br35H) with predefined splits for reliable training and testing. Methods included CNNs and Transformers, with paradigms like transfer learning and hybrid approaches. Optimization tools, such as the AdamW optimizer, improved regularization, while performance was assessed using accuracy, precision, recall, and F1-score. ViT achieved the highest accuracy (95.65%) but was resource-intensive, while MobileViT offered a practical balance of accuracy (90.39%) and efficiency. In ref. [116], the authors used CNN to classify brain tumors from MRI scans, integrating data augmentation and image processing for improved accuracy. The dataset, sourced from Kaggle, included 322 MRI images of healthy and tumor-affected brains. Preprocessing steps like noise removal, scaling, and grayscale conversion enhanced image quality, while FCM clustering segmented tumor regions. Features were extracted with DWT and reduced using PCA. The CNN model achieved a peak accuracy of 94.25% on the training set and 81.92% on validation, implemented using Google Colab with optimization tools such as NumPy and Pandas. This approach addresses challenges like noise and data imbalance, enabling early tumor detection. In ref. [124], the authors proposed a 3D GAN for brain tumor segmentation in 3D MRI images. The GAN integrates a generator and discriminator, combining FCNs, U-Net, and Residual Network architectures for improved segmentation. The generator creates segmented images, while the discriminator validates them against ground truth. Using the BraTS dataset, the network applied extended convolutions for feature extraction and a Dice coefficient-based loss function to balance foreground and background accuracy. Techniques like Group Normalization (GN), Spectral Normalization (SN), and adversarial training were employed for stability and enhanced performance. This approach addresses class imbalance and limited data, achieving precise segmentation, particularly for small tumor regions, and contributes to advancements in computer-aided medical diagnosis. In ref. [125] the authors compared various image segmentation methods for brain tumor detection in MRI images, including Otsu’s method, Watershed algorithm, Level Set method, K-Means clustering, DWT, and CNN. Simulations were conducted using the BraTS dataset. The study highlights CNN as the most effective technique, achieving the highest accuracy (91.39%) and fastest response time (2.519 s) among the methods evaluated. The CNN utilized a LeNet architecture, optimized for MRI segmentation, outperforming traditional approaches in metrics like accuracy, precision, recall, and F-measure. Pre-processing steps, such as noise reduction and histogram equalization, were applied across methods to enhance segmentation quality. In ref. [126], the authors developed a binary classification framework for diagnosing glioma brain tumors using T2 MRI brain images, integrating CNNs with DWT for improved feature extraction. By transforming MRI scans into the frequency domain, the approach captured critical spatial and temporal features often missed by traditional pixel intensity-based methods. The proposed CNN outperformed a conventional one, an SVM classifier, and a transfer learning model using a pre-trained VGG16 network, which was less effective for MRI-specific tasks. The study utilized datasets from the BraTS and ISLES challenges, along with anonymized scans from the Greek public General Hospital database, totaling 572 T2 MRI images from 572 patients, split into 382 for training and 190 for testing. With DWT-enhanced features, CNN achieved 97% accuracy, 100% sensitivity, 93% specificity, and 95% precision, significantly outperforming pixel-based models. The framework incorporated optimization techniques, such as dropout layers, RMSprop optimizer, and hyperparameter tuning, to enhance performance. In ref. [127], the authors introduced the Adaptive Feature Medical Segmentation Network (AFMS-Net), a deep learning paradigm for high-performance 3D brain lesion segmentation in medical imaging. The methodology leverages advanced CNN with novel encoder-decoder architectures, employing the Single Adaptive Encoder Block (SAEB) and Dual Adaptive Encoder Block (DAEB). These blocks integrate attention mechanisms, such as squeeze-and-excite and advanced spatial-channel attention, to refine feature extraction and enhance segmentation performance. The transfer learning AFMS-Net employs transfer learning paradigms by training on large public datasets, BraTs, ATLAS v2.0, and ISLES, highlighting its adaptability to various medical imaging scenarios. The segmentation framework further incorporates optimization strategies like the Adam optimizer and uses a custom loss function combining dice loss and categorical focal loss to address class imbalance issues and achieve precise segmentation. The authors utilize preprocessed datasets, including multi-modal MRI scans, ensuring homogeneity through techniques such as rigid registration and voxel resampling. In ref. [128], the authors proposed a hybrid deep learning framework for brain tumor classification using MRI. The methodology combines CNNs with sparse Autoencoders for data augmentation to address class imbalance in the Figshare dataset, which includes 3064 T1-WCEMRI images across three tumor classes: meningioma, glioma, and pituitary tumors. Two pretrained models, Inception ResNetV2 and EfficientNetB0, were fine-tuned with Bayesian Optimization to optimize hyperparameters like learning rate and momentum. Deep features extracted from the models were further refined using a Quantum Theory-based Marine Predator Algorithm (QTbMPA), designed to optimize feature selection for improved classification accuracy. The selected features were fused using a serial-based approach and classified using traditional and neural network-based classifiers, achieving a maximum accuracy of 99.80% with a sensitivity rate of 99.83% and a precision rate of 99.83%. In ref. [129], the authors applied CNNs with architectures such as VGG, ResNet, DenseNet, and SqueezeNet for diagnosing brain tumors from MRI images. They employed the paradigms of transfer learning and ensemble learning by using pre-trained models on the ImageNet dataset for feature extraction and then combining these features for classification. The study utilized a publicly available dataset of 7022 MRI images from the Kaggle platform, divided into glioma, malignant, pituitary, and no tumor classes. They optimized the CNN through hyperparameter tuning, using methods like grid search and Adaptive Moment Estimation (Adam) to enhance performance. The authors compared multiple machine learning classifiers, such as SVM, KNN, Decision Tree, and Naive Bayes, for feature classification and found that SVM with DenseNet features achieved the highest accuracy. Furthermore, ensemble learning combined the outputs of different CNN models to improve classification accuracy. ResNet, with optimized parameters, achieved a perfect accuracy of 100%, demonstrating the model’s efficacy in this domain. In ref. [130], the authors present a novel approach to brain tumor detection by integrating Proper Orthogonal Decomposition (POD) with CNNs and advanced transfer learning methods. The research applies paradigms such as rransfer learning and Exp XAI to MRI scans for binary classification of brain tumor presence. The authors utilized pre-trained deep learning models, including MobileNetV2, Inception-v3, ResNet101, and VGG-19, alongside the POD-CNN framework to achieve high accuracy and computational efficiency. The dataset employed was the BraTS dataset, comprising MRI scans labeled by expert radiologists, pre-processed with techniques like normalization, inhomogeneity correction, and data augmentation to enhance the data quality and mitigate biases. For optimization, hyperparameters such as learning rate, batch size, and dropout rates were fine-tuned, with models trained using the Adam optimizer. The POD-CNN method demonstrated efficiency by reducing computational costs while maintaining high accuracy (95.88%), though MobileNetV2 outperformed others with a 99.82% accuracy. XAI was addressed using SHapley Additive exPlanations (SHAP) to highlight the model’s interpretative insights into tumor boundary identification, enhancing trust in the predictions. In ref. [131], the authors propose CVG-Net, a novel hybrid neural network that combines 2DCNN and the pre-trained VGG16 model for transfer learning-based feature engineering to diagnose brain tumors using MRI scans. They employed a multi-class dataset of 21,672 MRI images categorized into glioma tumors, meningioma tumors, pituitary tumors, and normal samples. The proposed CVG-Net extracts spatial features from MRI images using 2DCNN and VGG16, creating a hybrid feature set that is then input into machine learning classifiers, including KNN, logistic regression, and random forest models. The study applies Synthetic Minority Oversampling Technique (SMOTE) to balance class distribution and conducts hyperparameter tuning to optimize performance. The methodology achieved state-of-the-art accuracy, with KNN achieving the best results of 96% accuracy under a 10-fold cross-validation setup. In ref. [132], the authors propose a deep transfer learning-based framework for brain tumor segmentation and classification using U-Net and a modified VGGNet. The process involves segmenting images with U-Net during pre-processing and classifying them with a transfer learning model. VGGNet’s final layers are modified to achieve high classification accuracy, leveraging pre-trained layers from ImageNet for feature extraction. The study uses a dataset of 3064 T1-WMRI scans from 233 patients, covering glioma, meningioma, and pituitary tumors. Optimization tools, including the Adam optimizer, categorical cross-entropy loss, dropout, and batch normalization, enhance performance. This approach achieves classification accuracies of 98.6%, 98.76%, and 99.45% for meningioma, glioma, and pituitary tumors, respectively, demonstrating the effectiveness of combining segmentation and transfer learning for medical image analysis. In ref. [133], the authors propose a method combining GRU networks with an enhanced Hybrid Dwarf Mongoose Optimization (EHDMO) algorithm to detect brain tumors from MRI images. GRUs process sequential data with low computational complexity, while EHDMO optimizes GRU parameters like weights, biases, and hidden units for improved accuracy. The method used the Brain-Tumor-Progression (BTP) dataset, featuring 8798 MRI images from 65 patients with glioblastoma and low-grade glioma, along with clinical metadata. Preprocessing included median filtering, data augmentation (geometric transformations, noise simulation), and image scaling. The approach achieved high performance metrics: sensitivity (0.98), specificity (0.97), accuracy (0.95), outperforming models like BrainMRNet, VGG19, and YOLOv2. In ref. [134], the authors applied a federated learning-based deep learning approach for brain tumor classification using MRI images. They utilized CNN, specifically a modified VGG16 architecture, to classify four tumor categories: glioma, meningioma, pituitary, and no tumor. The authors combined federated learning and transfer learning paradigms to address challenges like data privacy and the scarcity of annotated datasets. Federated learning allowed decentralized training across multiple clients without compromising patient confidentiality, while transfer learning leveraged pre-trained CNN weights from the ImageNet dataset to enhance classification performance. The model was trained on an integrated dataset comprising MRI images from figshare, SARTAJ, and Br35H datasets, with rigorous preprocessing steps like augmentation and normalization to standardize input data. The researchers incorporated optimization techniques such as dropout layers to prevent overfitting and hyperparameter tuning for enhanced performance. The model achieved an overall accuracy of 98%, with high precision, recall, and F1-scores across all categories, demonstrating its effectiveness in brain tumor diagnosis and its potential to improve clinical outcomes. In ref. [135], the authors developed Neuro-XAI, an explainable deep learning framework for brain tumor segmentation and classification using MRI scans. They employed DeepLabV3+ for segmentation with hyperparameters optimized via Bayesian methods and used transfer learning models (Darknet53 and MobileNetV2) for feature extraction, followed by Bayesian-optimized SVM for classification. The BraTS dataset, comprising various MRI modalities, was used for training and evaluation. XAI tools like Grad-CAM improved model interpretability, while the equilibrium optimizer enhanced feature selection. The framework achieved 98% segmentation accuracy and 97% classification accuracy, with uncertainty quantified using confusion entropy, offering a robust and interpretable solution for medical diagnostics. In ref. [136], the authors used transfer learning and IoT technologies to enhance brain tumor classification from medical images. Pre-trained CNNs like VGG-16 and ResNet were fine-tuned on medical datasets to extract hierarchical features, with data augmentation improving robustness and preventing overfitting. IoT integration enabled real-time data collection, enriching the training dataset. Evaluation using metrics such as accuracy, sensitivity, and F1 score showed superior performance over traditional methods like SVMs and random forests. Optimization techniques, including hyperparameter tuning and layer freezing, further refined the models, making the approach highly reliable for clinical diagnostics. In ref. [137], the authors applied CNNs for brain tumor classification using MRI imagery, leveraging a sequential architecture with convolutional layers for feature extraction, max-pooling for dimensionality reduction, dropout layers for regularization, and dense layers for final classification. They incorporated XAI through Grad-CAM visualizations to enhance interpretability. The dataset, sourced from figshare, SARTAJ, and Br35H repositories on Kaggle, included four classes: glioma, meningioma, pituitary tumor, and no tumor. Preprocessing involved resizing, normalization, and data augmentation (rotation, flipping, zooming). Optimization utilized the Adam optimizer and categorical cross-entropy loss function. Achieving 98% accuracy, with high precision, recall, and F1-scores, the model demonstrated strong efficacy and interpretability for brain tumor diagnosis. In ref. [138], the authors propose RF-ShCNN, a hybrid model combining Deep Residual Networks (DRN) and Shepherd CNN (ShCNN) for brain tumor detection using MRI images. Utilizing Transfer Learning, the method incorporates Adaptive Wiener Filtering (AWF) for noise reduction, CRF-RNN for segmentation, and feature extraction techniques like statistical measures, Gabor wavelets, GLCM, and DWT with LGXP. The hybrid model integrates a regression layer with Fractional Calculus (FC) for enhanced accuracy. Tested on the BraTS 2018 and Figshare datasets, it achieved 94% accuracy, 95% sensitivity, and 94.9% specificity, demonstrating its effectiveness in accurate tumor detection and classification. In ref. [139], the authors employed CNNs and kernel-based filtering for brain tumor diagnosis. The approach includes preprocessing MRI images using a hybrid probabilistic model for noise reduction and contrast enhancement, followed by kernel-based spectral pixel extraction and background filtering to estimate a refined covariance matrix. Segmentation was performed using a CNN trained to reconstruct brain regions, identifying tumor areas with maximum reconstruction loss through pixel-wise segmentation. The method was validated on the BraTS dataset, achieving an average Dice score of 0.82. Optimization techniques, including the Adam algorithm, were used to enhance performance, demonstrating high accuracy and robustness for tumors of various sizes.

Figure 12, generated from Table 5, showcases a comprehensive analysis of the methodologies, learning paradigms, and datasets utilized in brain tumor detection and classification studies. Figure 12a illustrates the variety of machine learning and deep learning methods, with CNN-based techniques being the most widely employed, alongside other approaches such as Transfer Learning, Grad-CAM, Ensemble Learning, GAN, UNet, and ViT, demonstrating the diversity of techniques used. Figure 12b focuses on the learning paradigms, highlighting Supervised Learning as the most prevalent, which includes techniques like Deep Learning, Sequential Learning, and Comparative Study. Transfer Learning is also extensively applied, with other paradigms such as Hybrid Approach, XAI, Adversarial Training, and Federated Learning contributing to the landscape. Figure 12c emphasizes the datasets, with BraTS being the most commonly used dataset, followed by others such as Kaggle, Private Datasets, and SARTAJ, along with specialized datasets like Greek Hospital Dataset and ATLAS v2.0. This figure provides a concise visualization of the key components in state-of-the-art brain tumor detection and classification studies.

## 6. Discussion and Pathways for Future Opportunities

The state-of-the-art in brain tumor detection using deep learning highlights significant advancements in methodologies, learning paradigms, and datasets, yet key gaps remain in fully leveraging these techniques. While CNNs and transfer learning have become dominant due to their efficiency in feature extraction, tumor segmentation, and classification, their reliance on large, labeled datasets still limits their applicability in diverse real-world settings. A critical gap exists in the generalizability of these models, particularly across different tumor types and patient populations. This issue is compounded by class overlapping, where tumor types exhibit similar features, leading to challenges in distinguishing them accurately. Moreover, while hybrid approaches have gained traction by combining various deep learning techniques for enhanced performance, these methods often lack the adaptability needed to address the complex, dynamic nature of brain tumor imaging. Custom architectures are promising but are frequently underutilized due to the challenge of tailoring solutions for specific datasets or tumor types.

Techniques such as Autoencoders, GANs, and LSTM networks remain less prevalent, despite their potential for addressing unique challenges such as data contextualization, temporal data modeling, and feature refinement. One reason for their underutilization could be the lack of sufficient research focused on their integration with brain tumor detection tasks. These methods could offer solutions to the complexities associated with imaging, such as noise reduction and the handling of varying tumor growth patterns over time. However, they often require more intricate setups and specialized knowledge to implement, which may contribute to their limited use.

Furthermore, datasets like BraTS, Kaggle, and T1-WCEMRI, while comprehensive and standardized, may not fully capture the variability seen in real clinical settings, where imaging techniques, tumor characteristics, and patient demographics can differ significantly. Personal clinical data present an exciting opportunity to bridge this gap, yet the challenges of data privacy, annotation quality, and clinical integration hinder its widespread adoption. Additionally, while supervised learning continues to dominate, there is a need for more research into unsupervised or semi-supervised methods, particularly as these could provide solutions in cases where labeled data are scarce. Finally, while emerging techniques such as ViT, federated learning, and adversarial training represent the field’s drive toward innovation, these advanced methods are often underutilized due to their complexity, computational demand, and the need for specialized infrastructure, further limiting their adoption in mainstream clinical applications.

Despite these hurdles, the continued exploration of these underutilized methods and the resolution of existing gaps offer significant opportunities for advancing brain tumor detection and classification. To drive progress in this field, we propose several future directions categorized into five key areas, as illustrated in the diagram presented in Figure 13.

### 6.1. Discussing and Improving Raw Image Quality

The quality of raw medical images significantly impacts the effectiveness of deep learning models in brain tumor detection [140,141,142,143,144,145,146]. Unfortunately, many studies treat available datasets as perfect representations without considering the effects of image type, acquisition protocols, or compression techniques on model performance [147,148,149]. Image compression, while essential for managing large datasets, often introduces artifacts or reduces critical diagnostic details. Methods like JPEG or JPEG2000 compression have not been adequately studied in this context, nor have adaptive compression techniques that can selectively retain vital image features while minimizing file size [147,148,149]. Furthermore, preprocessing methods aimed at enhancing image clarity and correcting for noise or motion artifacts are underutilized but have the potential to significantly improve data quality. Leveraging artificial intelligence-driven tools for dynamic image quality assessment and enhancement could ensure that inputs to deep learning models are consistent and reliable, even when acquired from diverse clinical settings [150,151]. Addressing these aspects will ensure that models trained on such data are more robust and applicable to real-world diagnostic scenarios.

### 6.2. Exploration of Underutilized Deep Learning Methods

While CNNs dominate brain tumor detection research, several promising methods remain underexplored. Transformers, for instance, are gaining traction for their ability to model complex relationships in data, making them suitable for handling spatial and temporal dependencies in medical imaging [152,153,154,155,156,157,158]. Graph Neural Networks (GNNs) offer another avenue, as they can represent and analyze relationships between imaging features, which is critical in understanding tumor morphology [159,160,161]. Autoencoders and GANs are also valuable tools for enhancing data augmentation, detecting anomalies, and enabling semi-supervised learning, especially when working with limited datasets. LSTM networks hold promise for capturing temporal changes in patient data, such as tracking tumor progression or response to therapy [162]. Moreover, hybrid approaches combining traditional CNNs with these advanced architectures could unlock new possibilities for improving feature extraction and decision-making processes. By integrating these methods, researchers can tackle unique challenges in brain tumor detection, such as small dataset sizes, complex tumor characteristics, and temporal dynamics.

### 6.3. Expanding and Diversifying Datasets

The reliance on a few public datasets, such as BraTS and Kaggle, has limited the diversity of data used for training and evaluating brain tumor detection models. Therefore, incorporating fewer common datasets, such as those from TCIA or MICCAI, can provide a broader representation of imaging conditions, tumor types, and patient demographics. Using private clinical datasets would further enhance the variability and realism of training data, ensuring models are better suited to real-world applications. Additionally, the integration of multimodal imaging data, such as combining MRI, CT, PET, and histopathological images, could enable the development of more robust and holistic diagnostic systems. Including non-imaging data, such as genomic, proteomic, and clinical records, could improve diagnostic precision by adding complementary insights into tumor behavior and patient-specific factors. Addressing underrepresented populations and imaging modalities is another crucial step toward creating more inclusive and generalizable models, ultimately enhancing the fairness and applicability of these systems.

### 6.4. Beyond Deep Learning: Model Behavior Integration

To push the boundaries of deep learning in brain tumor detection, it is essential to go beyond current architectures and incorporate model behavior analysis into the learning process. Techniques like recurrent expansion allow models to iteratively refine their parameters based on both data characteristics and the model’s dynamic responses during training [163,164,165,166]. These approaches enable a deeper understanding of feature mappings and how models behave under different training conditions, leading to more effective parameter tuning and improved performance. In recurrent expansion theories, small-scale machine learning techniques, such as ELMs, can complement deep learning by providing lightweight yet effective alternatives for feature extraction or ensemble learning [163,164,165,166]. Combining domain adaptation with transfer learning can further enhance model adaptability to diverse clinical scenarios, enabling models to handle variations in imaging modalities, patient demographics, and tumor characteristics more effectively.

### 6.5. Uncertainty Quantification

In clinical applications, particularly in predictions generated by deep learning models, quantifying uncertainty is as crucial as achieving high accuracy. Current brain tumor detection systems, as highlighted in the documented review presented in previous sections, often lack robust mechanisms for providing confidence estimates alongside predictions [167,168]. Techniques such as ensemble learning, confidence intervals, Bayesian deep learning, and Monte Carlo dropout, among others, can address this gap by enabling models to estimate uncertainty in their outputs [167,168]. These methods can be integrated into diagnostic pipelines to identify ambiguous cases that require further review, thereby improving clinical trust in artificial intelligence and learning systems. Additionally, XAI frameworks can enhance the interpretability of models, allowing clinicians to understand and validate predictions, fostering greater adoption of artificial intelligence in practice. Developing tools and methodologies for assessing uncertainty and integrating these into model performance benchmarks is critical for ensuring that artificial intelligence systems meet clinical reliability standards.

### 6.6. Additional Considerations

The development of standardized benchmarks aligned with specific evaluation methodologies is crucial to ensure consistency and comparability across studies in the field of tumor detection using deep learning. Additionally, integrating real-time and edge computing capabilities into detection systems could significantly enhance the deployment of these models in clinical settings, particularly in resource-constrained environments. In scenarios with limited connectivity or strict privacy requirements, federated learning emerges as a key solution, enabling collaborative model training while maintaining data privacy and addressing these challenges effectively. Lastly, fostering interdisciplinary collaboration among radiologists, computer scientists, and clinical researchers is vital for translating research innovations into practical diagnostic tools. These collaborative efforts will help bridge the gap between theoretical advancements and their application in real-world clinical scenarios, paving the way for more equitable, efficient, and accurate brain tumor detection systems.

## 7. Conclusions

In conclusion, this paper provided a comprehensive review of state-of-the-art studies in brain tumor detection using deep learning, encompassing over 100 references from the past half-decade. Through an in-depth analysis of related reviews, feature extraction techniques, segmentation methodologies, and classification approaches, it was observed that CNNs and transfer learning remain dominant methodologies across all aspects of tumor detection. However, innovative methods such as hybrid models, GANs, GNNs, and Transformers remain relatively underexplored despite their potential to address complex challenges in tumor detection. Similarly, learning paradigms like XAI and federated learning, which could enhance model interpretability and privacy-preserving collaborative training, have been sparsely discussed. Moreover, key aspects such as the impact of image types and compression, multimodal data integration, and uncertainty quantification have received insufficient attention, despite their critical role in improving model robustness and clinical applicability. The findings suggest that future research should prioritize addressing gaps related to data diversity by incorporating underutilized datasets and ensuring equitable representation of imaging modalities and patient demographics. Enhancing image quality through improved acquisition protocols, adaptive compression techniques, and artifact correction will be critical for maintaining diagnostic fidelity. Additionally, integrating multimodal imaging data with non-imaging data, such as genomic or clinical records, could enable the development of holistic diagnostic systems. Leveraging advanced technologies like federated learning and real-time edge computing would further facilitate the deployment of deep learning models in clinical settings, particularly in resource-constrained environments. By addressing these gaps, future advancements can bridge the divide between theoretical research and practical clinical application, leading to more robust, interpretable, and accessible diagnostic systems. This review provides valuable insights into the current landscape and identifies potential directions for advancing brain tumor detection with deep learning, offering a foundational reference for researchers and practitioners striving to improve patient outcomes. Finally, this review explores methods, paradigms, and datasets for feature extraction, segmentation, and classification, leaving a detailed comparative analysis of performance metrics, such as accuracy, as a valuable direction for future studies to better assess methodological effectiveness.

## Figures and Tables

**Figure 1 jimaging-11-00002-f001:**
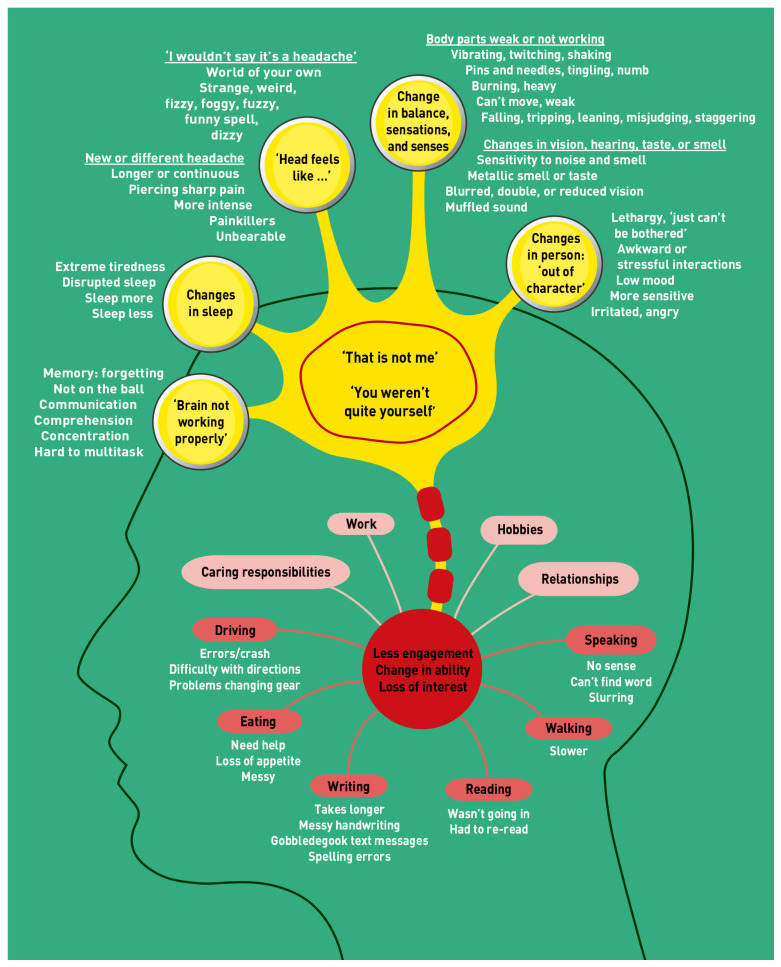
Illustration of changes or symptoms prior to brain tumor diagnosis. This figure, reproduced from [14] under an open-access license permitting non-commercial use, has been edited using a curves adjustment layer. This adjustment was applied to manipulate the visual properties, enhancing clarity and detail.

**Figure 2 jimaging-11-00002-f002:**
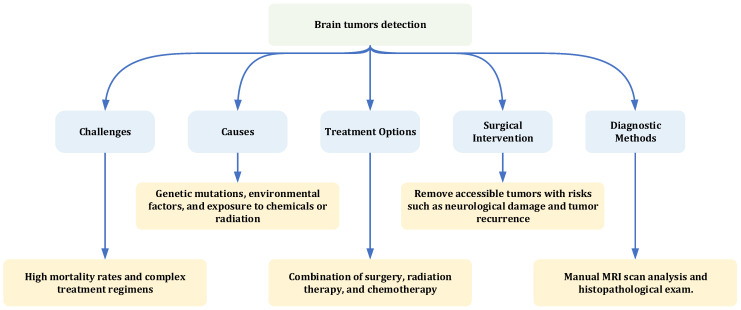
Key aspects of brain tumors: challenges, causes, treatments, and diagnostic methods.

**Figure 3 jimaging-11-00002-f003:**
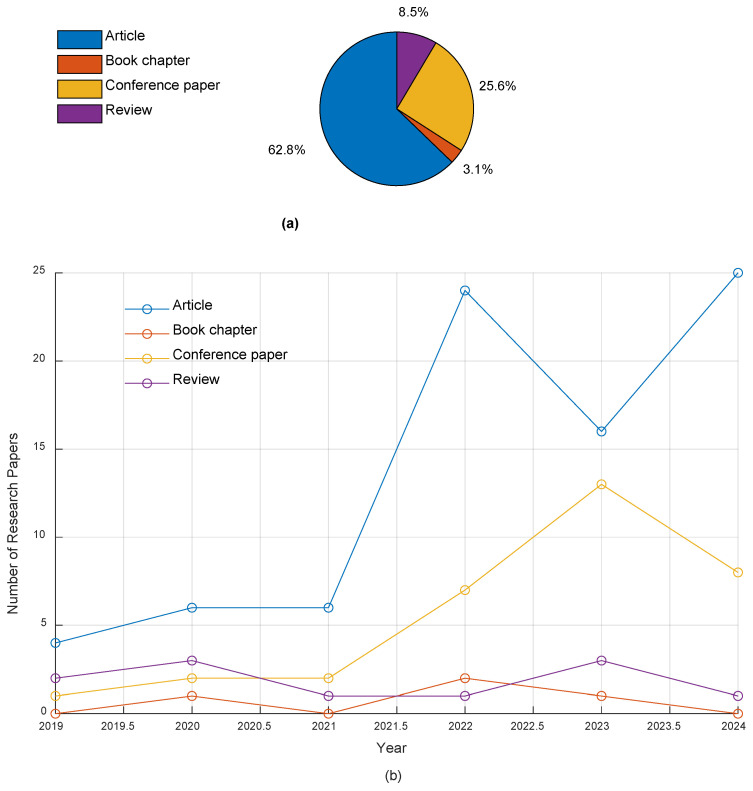
Distribution of research papers by type: (**a**) pie chart showing the percentage of publications by type over half a decade; (**b**) number of research papers per year by type.

**Figure 4 jimaging-11-00002-f004:**
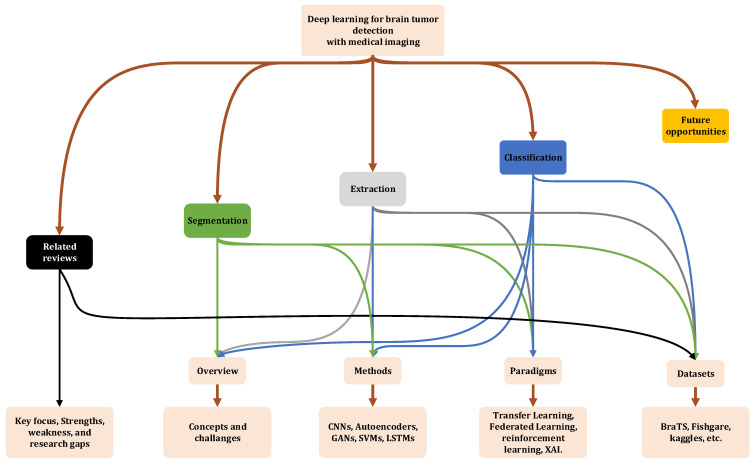
Simplified diagram of the papers framework for reviewing brain tumor detection related works: incorporating related review analyses, feature extraction, segmentation, and classification using medical images and deep learning, with future opportunities.

**Figure 5 jimaging-11-00002-f005:**
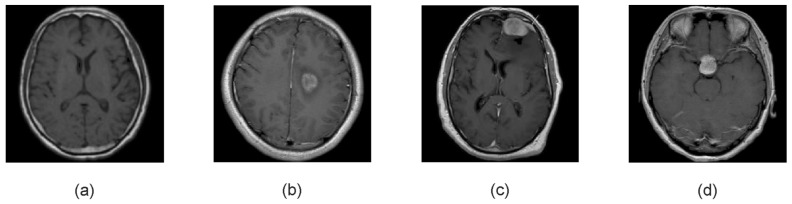
Examples of healthy and cancerous brain images: (**a**) healthy brain; (**b**) glioma tumor; (**c**) meningioma tumor; (**d**) pituitary tumor.

**Figure 6 jimaging-11-00002-f006:**
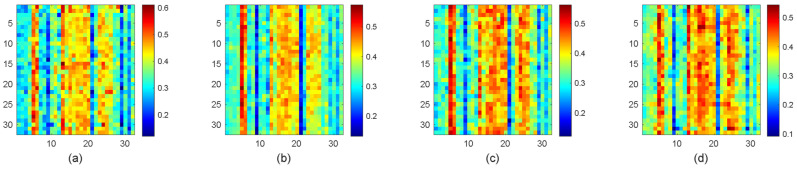
Examples of extracted features with an autoencoder from healthy and cancerous brain images: (**a**) healthy brain; (**b**) glioma tumor; (**c**) meningioma tumor; (**d**) pituitary tumor.

**Figure 7 jimaging-11-00002-f007:**
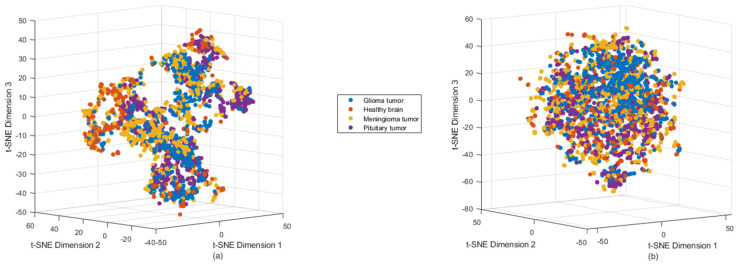
Examples of data scatters of (**a**) original data and (**b**) extracted features with an autoencoder from healthy and cancerous brain images.

**Figure 8 jimaging-11-00002-f008:**
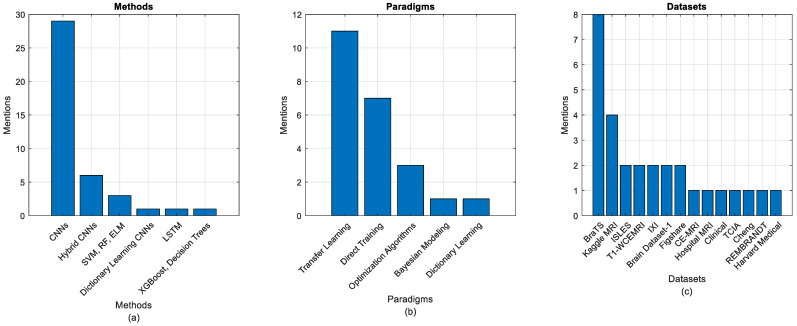
(**a**–**c**) Summary of feature extraction methods, paradigms, and datasets.

**Figure 9 jimaging-11-00002-f009:**
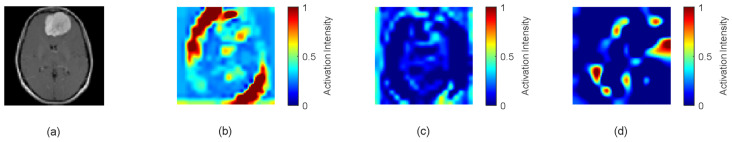
Example of MRI brain image segmentation: (**a**) original MRI brain image; (**b**–**d**) ResNet-18 activations at layers 5, 10, and 20 highlighting key segmentation areas.

**Figure 10 jimaging-11-00002-f010:**
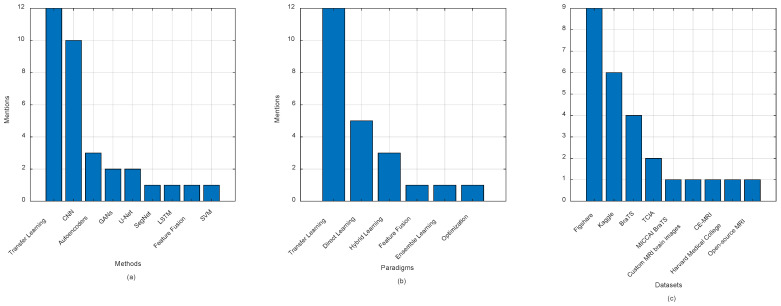
Overview and distribution of methods used for brain MRI image segmentation: (**a**) bar chart depicting the frequency of methods employed; (**b**) bar chart categorizing the learning paradigms; (**c**) bar chart illustrating the distribution of datasets used.

**Figure 11 jimaging-11-00002-f011:**
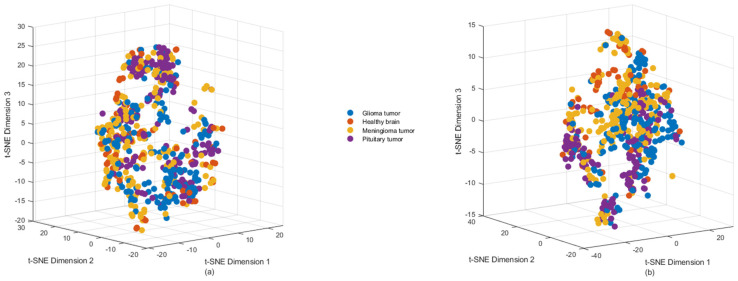
Visualization of data distributions from validation images and ResNet mappings: (**a**) scatter plot of feature distributions extracted directly from the original validation images; (**b**) scatter plot of feature distributions obtained from ResNet layer activations, illustrating the network’s learned representations.

**Figure 12 jimaging-11-00002-f012:**
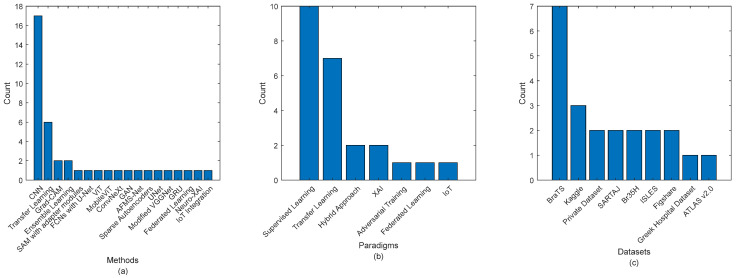
Comprehensive analysis of brain tumor detection and classification techniques: (**a**) methods; (**b**) learning paradigms; and (**c**) datasets.

**Figure 13 jimaging-11-00002-f013:**
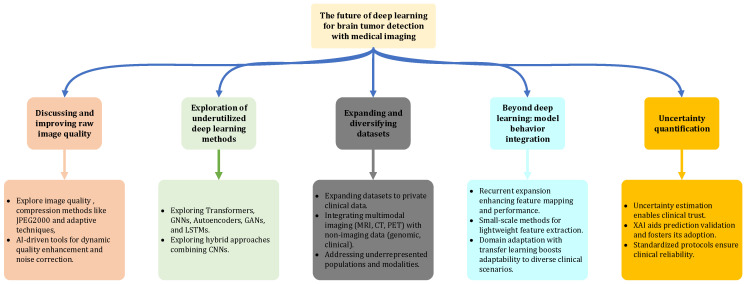
Diagram of proposed future directions for brain tumors detection with deep learning image classification.

**Table 1 jimaging-11-00002-t001:** Analysis criteria for brain tumor detection methods.

Aspect	Segmentations	Feature Extraction	Classifications
Applications	Pixel-wise classification and boundary delineation of tumors, especially in MRI scans.	Extraction of significant features from medical images to aid in tumor detection and analysis.	Categorization of brain tumors as benign, malignant, or other types based on medical images.
Methods	CNNs, LSTMs, U-Net, Autoencoders, GANs, etc.		
Paradigms	Transfer learning, reinforcement learning, federated learning, supervised learning, XAI, IAI, etc.		
Descriptions of some methods	CNNs: Used for pixel-wise tumor extraction, segmentation, and classification in medical images. U-Net: Specialized CNNs for biomedical segmentation. LSTMs: Capture temporal dependencies. Autoencoders: Denoise images extract key features for tumor segmentation and retain essential image features. GANs: Generate synthetic data for enhanced feature extraction.		
Description of Some paradigms	Transfer learning: Adapts pre-trained models for tumor segmentation, enhancing accuracy. Federated learning: Allows distributed learning while protecting data privacy. Reinforcement learning: Improves models over time. XAI: Provides transparency in model decisions.		

**Table 2 jimaging-11-00002-t002:** Summary of related review papers analyses.

Ref.	Key Focus	Datasets	Strengths	Research Gaps
[28]	Deep learning; Glioma detection; MRI images; CNNs and U-Nets	BraTS, TCIA, IXI, ISLES	Discussion on CNNs, U-Net, and transfer learning; advantages of deep learning in automatic feature extraction; explores pre-processing, model architectures, and evaluation criteria; offers a global analysis of methods.	Limited research on glioma classification and feature extraction; underuse of advanced models like GANs, transformers, and CNN-LSTM hybrids. Insufficient focus on brain tumor types other than glioma.
[29]	Brain tumor segmentation; MRI; analysis range (2019–2023).	BraTS, TCIA (T1, T2, FLAIR MRI modalities).	Non-AI, machine learning, deep learning, hybrid; current techniques and their performance.	Lack of detailed analysis on integration with clinical practices, limited exploration of multimodal data beyond MRI, insufficient discussion on emerging lightweight models.
[30]	Investigation of machine learning and deep learning algorithms for brain tumor detection, focusing on segmentation and classification using MRI.	BraTS dataset prominently referenced.	Systematic breakdown of methods, performance metrics, and methodologies; detailed analysis of CNNs for segmentation and classification.	Opportunity to enhance feature extraction techniques, improve segmentation precision for larger datasets, and explore methods for better segmentation of whole, core, and enhancing tumor regions.
[31]	Image segmentation methods for MRI-based brain tumor detection using machine learning, comparing traditional and deep learning models.	BraTS challenge datasets prominently discussed.	Comprehensive analysis spanning two decades, comparison of traditional and deep learning methods, highlights deep learning’s superior performance.	Need for integrating feature engineering with segmentation to enhance model robustness and accuracy, exploring methods for better differentiation between tumor subtypes or stages.
[32]	Machine learning and deep learning techniques for brain tumor analysis in MRI; classification, segmentation, and feature extraction.	MRI datasets focused on glioma, meningioma, and pituitary tumors	Detailed analysis of feature extraction and CNN-based models, highlights trade-offs between accuracy, complexity, and efficiency	Insufficient evaluation of non-CNN models and emerging paradigms; limited discussion on multi-modality approaches and their impact on diagnostic improvements
[33]	Techniques for brain tumor segmentation using MRI, covering supervised and unsupervised methods.	BraTS, IBSR	Comprehensive coverage from classical to advanced machine learning methods, detailed analysis of CNNs, and emphasis on large datasets.	Limited focus on hybrid techniques, unsupervised learning, real-time segmentation (e.g., autoencoders), and advanced architectures like GANs; insufficient discussion on integration across tasks.
[34]	Deep learning techniques for MRI brain tumor analysis, focusing on classification and segmentation	BraTS, TCIA (TCGA-GBM, TCGA-LGG)	Structured analysis, clear strengths and limitations, discusses pretrained models (U-Net, VGG, AlexNet, ResNet), covers advanced topics like transfer learning and data augmentation	Need for discussions on model interpretability, improved solutions for small datasets, more focus on classification and feature extraction
[35]	Artificial intelligence methods for brain tumor segmentation and classification using MRI; review of over 100 papers	BraTS, TCIA (T1, T1c, T2, FLAIR MRI modalities)	Review of over 100 research papers, highlighting deep learning methods like CNNs, LSTMs, RNNs, and GANs.	Does not address extraction and classification as much as segmentation.
[36]	Deep learning applications in brain tumor analysis; segmentation and classification; challenges, limitations, and future research directions.	BraTS, TCGA-GBM, TCGA-LGG	Comprehensive review of segmentation and classification techniques; Discussion of 2D and 3D CNNs, FCNs, and their applications.	Insufficient exploration of feature extraction techniques relative to segmentation and classification. Limited analysis of emerging hybrid models that integrate feature extraction with classification tasks.

**Table 3 jimaging-11-00002-t003:** Overview of state-of-the-art deep learning for brain tumor feature extraction: techniques, paradigms, features, and performance on key datasets.

Study	Methods	Paradigms	Extracted Features	Advantages	Limitations	Dataset
[46]	XGBoost, SVM, Decision Trees, FCNN	Direct training	Radiomic features: First-order statistics, Shape-based, Gray Level Co-occurrence Matrix, Gray Level Run Length Matrix, Local Binary Patterns	High accuracy with XGBoost (88.51%) and FCNN (87.09%); effective feature extraction and analysis methods	Challenges in segmenting MRI modalities; dataset variability and some segmentation flaws in noise removal	Kaggle dataset (3265 MRI images: glioma, meningioma, pituitary tumor, healthy cases)
[59]	AlexNet, GoogleNet	Transfer learning	Morphological operations, CNN feature mapping	Accurate brain tumor segmentation and classification	Relies on pre-trained models, may require extensive fine-tuning	BraTS (2013–2017), ISLES
[60]	VGG19 CNN	Transfer Learning, Block-wise Fine-tuning	Low-level (edges), high-level (shapes, tumor-specific features)	Reduces overfitting, leverages domain-specific adaptations	Limited to MRI-only data, may not generalize across modalities	CE-MRI
[61]	CNN GLCM CA	Direct training	Texture features: ASME, Entropy, refined by CA	Effective in detecting texture details, accurate segmentation	Limited focus on CNN optimization and transfer learning	Hospital MRI dataset
[62]	ARKFCM Hybrid GLCM, LBP, HOG	Genetic Algorithm based selection	Texture and gradient-based features	High classification accuracy, hybrid approach enhances feature selection	Computationally expensive, hybrid model complexity	T1-WCEMRI
[63]	CNN Semantic Segmentation	Transfer Learning, SGDM Optimization	Tumor segmentation with semantic overlays	High classification accuracy (>99%)	Requires optimization settings, specific training configurations	T1-WCEMRI from 233 patients
[64]	CropNet	Direct training	Metastatic region extraction from candidate ROI	High sensitivity for small metastases, robust data augmentation	Custom architecture may require domain-specific tuning	217 MRI scans (158 patients)
[65]	Custom CNN, Transfer Learning (VGG-16, ResNet-50, Inception-v3)	Transfer Learning.	Image patterns, domain-specific features	High training accuracy (100%), effective with limited data	May overfit with smaller datasets	Kaggle MRI dataset (253 images)
[66]	Deep Autoencoder BFC	Jaya Optimization Algorithm	Wavelet entropy, scattering transform, information-theoretic features	Robust feature representation, effective for segmentation	May be computationally intensive	BraTS
[67]	Transfer Learning, ResNet-50 + Grad-CAM	Transfer learning	Heatmap localization, interpretable features	High interpretability, confidence-based classification	Limited dataset size, domain-specific constraints	IXI, ECKC, TCIA
[68]	Bi-directional MGLCM and LSTM	Direct training	Texture features, brain symmetry	High classification accuracy, optimized for symmetry analysis	Does not use advanced feature engineering techniques	Clinical dataset (214 MRI) and BraTS
[69]	CNN and DWA	Direct training	Spatial and segmented features	High accuracy (98%), robust for early detection	Limited to specific dataset types, specialized training	BraTS, ISLES
[70]	CDLLC and Local Constraint	Transfer learning	Discriminative feature representation	Strong performance, effective dictionary learning	Computational demands for dictionary learning	Cheng, REMBRANDT
[71]	GMCNN and Transfer learning	Transfer Learning, LOPO cross-validation	Spatial, orientation-specific features	High efficiency, robust cross-validation	Lacks inter-study comparability due to specialized features	BraTS
[73]	Hahn-PCNN-CNN (GoogLeNet + Hahn Moments)	Transfer Learning, Multi-modal Fusion	Complex textures, metabolic information	Enhances diagnostic utility, multi-modal fusion benefits	High computational complexity	Harvard Medical School (8000 images)
[74]	CNN + SVM, RF, ELM	Direct training	Region proposals, tumor localization	High segmentation accuracy, multiple classifier options	May require extensive training data, model complexity	Kaggle, Figshare
[75]	Hybrid CNN + SVM-RBF, RF, ELM	Direct training	CNN features, region proposals via RPNs	High classification accuracy (98.3% on Kaggle dataset, 98.0% on Figshare), precise tumor localization	May require optimization of region proposal network settings	Kaggle, Figshare
[76]	Transfer learning (ResNet-152) and GLCM	Transfer learning, COVID-19 Optimization Algorithm	Texture features: contrast, energy, correlation, homogeneity, entropy	Enhanced accuracy, reduced computational complexity	Dependence on GLCM feature consistency	BraTS
[77]	CNN, AFDF) and FCM	ACV-DHOA Optimization	Combined CNN and FCM features, adaptive clustering	Optimized segmentation and classification, ensemble-based accuracy	Complex optimization scheme	Kaggle MRI dataset (253 images)
[78]	BDWCNN	Direct training	Tumor-specific spatial features	High efficiency, reduced computational cost, robust metrics	Requires specific Bayesian parameter settings	BraTS, IXI

**Table 4 jimaging-11-00002-t004:** Comparative analysis of brain tumor segmentation and classification methods.

Ref.	Segmentation Methods	Paradigm	Advantages	Limitations	Dataset(s) Used
[91]	Variational Autoencoders (VAEs) + GANs	Transfer Learning	Enhances dataset by generating high-quality synthetic MRI images, avoids mode collapse	Does not explicitly perform segmentation, focuses on data generation	Figshare brain tumor MRI dataset (3064 images)
[92]	CNN (Convolutional Neural Network)	Direct Learning	Effective feature extraction, preprocessing enhances image quality for segmentation	Limited to tumor region identification, requires extensive preprocessing	MRI dataset (multiple tumor types)
[93]	AlexNet, VGGNet with CNN	Transfer Learning	Combines strengths of different architectures, high classification accuracy	High computational demand due to parallel network usage	Figshare brain tumor dataset, TCIA
[94]	SegNet + GLCM features	Transfer Learning	Reduced computational complexity by focusing on tumor-relevant areas	Limited generalizability to other tumor types, uses handcrafted features	BraTS (multi-modal MRI scans)
[95]	U-Net with Inception Modules	Transfer Learning	Multi-scale feature capturing, improved segmentation accuracy	Complexity increases with hybrid architecture	MICCAI BraTS dataset
[96]	Pretrained VGG-16, ResNet-50, Inception-v3	Transfer Learning	High accuracy with less data, efficient for MRI-specific data	Model performance heavily reliant on transfer learning, large model size	Kaggle (253 MRI images)
[97]	Adaptive Histogram Contrast Normalization, Otsu thresholding	Direct Learning	Simplicity, avoids complex deep learning architectures	Limited segmentation precision, no advanced architectures used	Custom MRI brain images
[98]	CNN + LSTM	Hybrid Learning	Captures spatial and temporal dependencies, high accuracy	Limited to binary classification, dependent on dataset quality	Kaggle (253 MRI images)
[99]	Graph Attention Autoencoder + CNN	Hybrid Learning	Combines structured and unstructured data, robust classification	Computationally expensive, may require large memory	Kaggle MRI datasets (multiple classifications)
[100]	Custom 17-layer CNN + MobileNetV2 + SVM	Transfer Learning	Entropy-based feature selection improves feature optimization	High computational demand due to custom and hybrid architecture	BraTS 2018, Figshare MRI datasets
[101]	CWCSO-enabled CNN with Fractional Probabilistic Fuzzy Clustering	Transfer Learning + Optimization	High accuracy and sensitivity in MRI segmentation	Complex optimization process, potential for overfitting	BraTS dataset
[102]	Median filtering, morphological operations, GLCM features	Feature Fusion	Combines texture and deep features, improved classification accuracy	Limited to certain image characteristics, handcrafted features needed	Figshare dataset
[103]	VGG16 Ensemble, Machine Learning Techniques	Ensemble Learning	Combines classifiers for enhanced accuracy, prevents overfitting with early stopping	Model complexity, requires ensemble tuning	Public MRI datasets (3787 images)
[104]	CNN with image enhancement techniques (Gaussian blur, CLAHE)	Direct Learning	Enhances image quality, increases model specificity	Limited robustness to dataset variability, complex preprocessing	Figshare, SARTAJ, BR35H datasets
[105]	YOLOv7-based CNN with CBAM, SPPF+, and BiFPN	Transfer learning, initialized with COCO pre-trained weights	Effective classification of gliomas, meningiomas, and pituitary tumors; improved feature extraction for varying tumor sizes	Requires high computational resources for complex CNN components	Open-source MRI dataset from Kaggle (10,288 images)
[106]	CNN with political optimizer	CNN optimized by enhanced political optimizer (opposition-based learning and chaos theory)	High classification accuracy (96%) with refined CNN architecture; robust against data variability	Computational cost of optimizer; possible overfitting without careful parameter tuning	Figshare brain tumor dataset
[107]	Feature Enhanced Stacked Auto Encoder (FESAE) with DWT and RGB channelization	Stacked Autoencoder with feature enhancement	High accuracy (98.61%) with enhanced feature details from spatial and frequency domains	Limited interpretability of stacked autoencoder features	MRI dataset from Kaggle and Harvard Medical College (2000 images)
[108]	Multi-Modal Generative Adversarial Network (MMGAN) with deep residual U-Net	GANs with residual networks and attention mechanisms	Superior segmentation accuracy and stability in training; prioritization of valuable feature channels	Requires large training datasets; adversarial training can be unstable	BraTS dataset (T1, T1c, T2, FLAIR modalities)
[109],	EfficientNet (B0-B4) with transfer learning	Transfer learning with pre-trained ImageNet weights	High accuracy, precision, and recall for tumor classification; visual interpretability with Grad-CAM	May require fine-tuning for new datasets; risk of overfitting	CE-MRI brain tumor dataset from Figshare
[110]	Modified AlexNet for noninvasive glioma grading	CNN with hyperparameter tuning	Accurate noninvasive classification; effective as an alternative to biopsy	Limited generalizability across different glioma grades and datasets	TCIA dataset (FLAIR-weighted MR images, 4069 slices

**Table 5 jimaging-11-00002-t005:** State-of-the-art analysis of brain tumor detection and classification techniques.

Study	Methods	Paradigms	Extracted Features	Advantages	Limitations	Dataset
[111]	SAM with adapter modules	Transfer learning	Mask decoder, SAM backbone	High Dice score (87.33%), efficient computational requirements	Private dataset limits reproducibility	Private dataset (489 multimodality MRI images)
[112]	CNN	Supervised learning	Convolutional features	High validation accuracy (97%), effective for binary classification	Limited dataset diversity	Kaggle dataset (3060 MRI images)
[113]	CNNs, Transfer Learning, Ensemble Learning	Deep learning	Pre-trained model features, ensemble features	High classification accuracy (98%)	Resource-intensive ensemble models	Dataset (3200 MRI images, 4 classes)
[114]	FCNs with U-Net	Deep learning	Multi-contrast MRI features, hierarchical segmentation	High Dice coefficient, robust segmentation	Complex multi-stage architecture	BraTS dataset
[115]	ViT, MobileViT, ConvNeXt	Hybrid approach	Transformer and CNN features	ViT achieves 95.65% accuracy	High resource requirements for ViT	SARTAJ, Br35H datasets
[116]	CNN	Supervised learning	DWT, PCA for dimensionality reduction	Noise and data imbalance addressed	Lower validation accuracy (81.92%)	Kaggle dataset (322 MRI images)
[124]	3D GAN	Adversarial training	3D MRI segmentation features	Precise segmentation, handles class imbalance	GAN training complexity	BraTS dataset
[125]	CNN	Supervised learning	Statistical and deep features	CNN outperformed traditional methods	MATLAB simulation limits scalability	BraTS dataset
[126]	CNN with DWT	Deep learning	Frequency domain features	High sensitivity and specificity	Smaller dataset size	BraTS, ISLES, and Greek hospital datasets
[127]	AFMS-Net	Transfer learning	Advanced encoder-decoder features	Addresses class imbalance, high Dice coefficient	Complexity in implementation	BraTS, ATLAS v2.0, ISLES datasets
[128]	CNN with Sparse Autoencoders	Transfer learning	Deep features fused using QTbMPA	Achieved 99.80% accuracy	Computational complexity	Figshare dataset (3064 images)
[129]	Ensemble of CNNs	Transfer learning	Features from ImageNet models	Perfect accuracy (100%) with ResNet	High computational cost	Kaggle dataset (7022 MRI images)
[130]	POD-CNN	Transfer learning, XAI	Spatial features, SHAP-based insights	High efficiency and interpretability	Limited to binary classification	BraTS dataset
[131]	CVG-Net (2D CNN + VGG16)	Hybrid learning	Hybrid spatial features	High accuracy (96%), balanced class distribution	Dependent on pre-trained models	Multi-class dataset (21,672 MRI images)
[132]	UNet, Modified VGGNet	Deep learning	Segmentation and classification features	High accuracy across classes	Focused on segmentation + classification	Dataset (3064 T1-weighted MRI scans)
[133]	GRU + EHDMO	Sequential learning	Temporal and optimized features	Outperforms other models, high specificity	Requires clinical metadata	Brain-Tumor-Progression dataset (8798 images)
[134],	Federated Learning + CNN	Transfer learning	Pre-trained weights for feature extraction	Ensures data privacy, 98% accuracy	Dependent on federated infrastructure	Integrated dataset (Figshare, SARTAJ, Br35H)
[135]	Neuro-XAI (DeepLabV3+, MobileNetV2)	XAI	Grad-CAM insights, equilibrium-optimized features	Explainable model, high accuracy	Relatively complex framework	BraTS dataset
[136]	CNN with IoT Integration	Transfer learning, IoT	Hierarchical features	Real-time data processing, high reliability	Infrastructure-dependent	Medical datasets (unspecified)
[137]	CNN with Grad-CAM	Deep learning	Grad-CAM-based interpretability	High accuracy (98%), interpretable predictions	No focus on segmentation	Figshare, SARTAJ, Br35H datasets
[138]	RF-ShCNN	Hybrid learning	Statistical and deep hybrid features	Accurate segmentation and detection	Complexity in hybrid design	BraTS 2018, Figshare datasets
[139]	CNN with kernel-based filtering	Supervised learning	Spectral pixel-based features	Robust tumor detection	Moderate Dice score (0.82)	BraTS dataset

## Data Availability

As part of our commitment to scientific advancement, we are sharing the data on downloaded papers along with the corresponding code in this link here: https://doi.org/10.5281/zenodo.14551069. Additionally, the codes for the examples discussed in this review, covering extraction, segmentation, and classification, are available in this link here: https://doi.org/10.5281/zenodo.14551091.

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
