# Peer review of "The Neural Frontier of Future Medical Imaging: A Review of Deep Learning for Brain Tumor Detection"

_2313-433X, 2024, doi:10.3390/jimaging11010002_

Round 1
Reviewer 1 Report
Comments and Suggestions for Authors
The manuscript presents a review study of deep learning for brain tumor detection based on more than 100 research articles from 2019 to 2024. Research articles are analyzed from the perspectives of feature extraction, segmentation and classification. Findings are thoroughly presented and discussed. It is believed that readers of the manuscript can get valuable insights into this research field.
Some minor suggestions and comments are presented in the following for the author to refine the manuscript.
It seems that some articles from IEEE Access are not included in the review. Is it due to the usage of the software "Publish or Perish" mentioned in the manuscript? Will it cause the incomplete collection of research articles? If some criteria are applied in the collection (filtering) process, it is recommended that the author can mention them in the manuscript.
Some articles do not contain the term "brain" and "tumor" in the title. For instance, the article "A Systematic Review on Recent Advancements in Deep Learning and Mathematical Modeling for Efficient Detection of Glioblastoma" in IEEE Transactions on Instrumentation and Measurement is not collected for the review.
Please provide the search expression used in the collection process. Readers might want to reuse the expression to investigate related research articles in the future.
According to the manuscript, 55 journal articles and 22 conference papers are downloaded. It means some articles are excluded. Please elaborate on the reason behind the filtering process.
It is recommended that the term "flowchart" used in the manuscript (e.g., Figure 4 and Figure 13) can be revised.
The manuscript presents comprehensive reviews and detailed introduction to research papers. It is highly recommended that the organization of the manuscript can be refined (e.g., divide long paragraphs into sections based on research papers) to achieve better readability.
Please check the term "Pradigms" in Figure 10.
Please revise the section numbering (Section 6.6 is missing).
Author Response
Comment 1: The manuscript presents a review study of deep learning for brain tumor detection based on more than 100 research articles from 2019 to 2024. Research articles are analyzed from the perspectives of feature extraction, segmentation, and classification. Findings are thoroughly presented and discussed. It is believed that readers of the manuscript can get valuable insights into this research field.
Response 1: Thank you for your thoughtful comments and kind feedback on our paper. We have made revisions based on your suggestions to the best of our ability, and we sincerely hope that the revised version will meet your expectations and receive your recommendation for acceptance.
Comment 2: It seems that some articles from IEEE Access are not included in the review. Is it due to the usage of the software "Publish or Perish" mentioned in the manuscript? Will it cause the incomplete collection of research articles? If some criteria are applied in the collection (filtering) process, it is recommended that the author can mention them in the manuscript.
Response 2: Thank you for your valuable feedback. The following statement has been added to Section 1.1, just before Figure 3, to clarify this gap in the methodology.
“It should be noted that the author did not intentionally exclude papers from any specific publisher during the paper collection process, which was based on the aforementioned criteria. As stated, the filtering process was carried out using a program script designed to minimize human error, such as overlooking certain papers or introducing biases due to lapses in focus. While it is possible that some articles were unintentionally excluded, it is guaranteed that the selected papers are directly relevant to the topic of deep learning and brain tumor detection. Although no methodology is perfect and some exclusions may have occurred, the quantity and quality of the selected papers are deemed sufficient to draw generalized conclusions.”
Comment 3: Some articles do not contain the term "brain" and "tumor" in the title. For instance, the article "A Systematic Review on Recent Advancements in Deep Learning and Mathematical Modeling for Efficient Detection of Glioblastoma" in IEEE Transactions on Instrumentation and Measurement is not collected for the review.
Response 3: Thank you for your valuable comment. It is possible that some articles, such as the mentioned paper, were unintentionally excluded because the script used for paper collection may not have captured terms like "brain" or "tumor" in the title, depending on the specific search criteria applied. Additionally, there were technical challenges during the collection process, including issues related to access restrictions, such as not being subscribed to certain journals or content being behind paywalls, which made it difficult to include all potentially relevant papers. However, we appreciate your reference, and we have cited the mentioned paper in the introduction section, in ref [27], right after the paragraph following Figure 2. We believe that the statement in response to comment 2 clarifies this issue properly. Despite these limitations, the papers included in the review are highly relevant to the topic, and we believe that the quantity and quality of the selected articles are sufficient to draw generalized conclusions.
Comment 4: Please provide the search expression used in the collection process. Readers might want to reuse the expression to investigate related research articles in the future.
Response 4: Thank you for your valuable comment. The search expression used in the collection process is included in Section 1.1 in the first few lines, where all the details about how we collected the papers are provided. Additionally, I am committed to posting the links to the script used in this expression, along with the collected data, in the data availability statement upon acceptance of the paper. This will not only allow readers to understand the collection process but also enable them to reuse and improve it for future use.
Comment 5: According to the manuscript, 55 journal articles and 22 conference papers are downloaded. It means some articles are excluded. Please elaborate on the reason behind the filtering process.
Response 5: Thank you for your comment. The filtering process was not intended to exclude any relevant articles, and it was conducted using a program script designed to minimize human error and biases. The script focused on gathering papers directly related to deep learning and brain tumor detection, but it is possible that some articles were unintentionally excluded due to the search criteria, such as the absence of specific terms like "brain" or "tumor" in the title. Additionally, some content was not included due to access restrictions, as certain papers were behind paywalls or not open access, which made it difficult to collect all potentially relevant research. However, we believe the papers included in the review are highly relevant to the topic, and the quantity and quality of the selected articles are sufficient to draw generalized conclusions. Thank you for your comment. The first part of your question regarding the filtering process has already been addressed in response to Comment 2. As for the second part, regarding challenges related to non-open access content, this was not explicitly mentioned or clarified in the manuscript, as I believe it may not be appropriate to include such details in the paper. However, I appreciate the importance of understanding the challenges faced during the collection process, and I hope the explanation provided in the response to Comment 2 clarifies the filtering methodology used.
Comment 6: It is recommended that the term "flowchart" used in the manuscript (e.g., Figure 4 and Figure 13) can be revised.
Response 6: Thank you for your suggestion. The term "flowchart" has been replaced with "diagram" in the manuscript, including in Figure 4 and Figure 13, as per your recommendation.
Comment 7: The manuscript presents comprehensive reviews and detailed introduction to research papers. It is highly recommended that the organization of the manuscript can be refined (e.g., divide long paragraphs into sections based on research papers) to achieve better readability.
Response 7: Thank you for your valuable feedback. The entire text has been reviewed, and long paragraphs have been separated into smaller ones. Please check the entire text, where the beginning of each newly separated paragraph is highlighted in red. Additionally, the following statement has been added to Section 1.3, after Figure 4, in response to your comment.
“In this architecture of the current review paper, it is acknowledged that the long paragraphs discussing the papers may be difficult to read. To address this, we have made efforts to improve readability by introducing subsections. However, we believe the current subsectioning is the most suitable in this case, as it allows for clear presentation of the review results in tables, followed by bar charts. Furthermore, the review is structured around three major axes: classification, extraction, and segmentation. We believe that the inclusion of bar charts and diagrams will greatly assist readers in understanding the recent trends in each subfield and provide valuable insights for drawing meaningful conclusions. We trust that this organization will enhance the clarity of the manuscript while ensuring a comprehensive review of the topic.”
Comment 8: Please check the term "Pradigms" in Figure 10.
Response 8: Thank you for your feedback. We have revised Figure 10 and similar figures to address this issue.
Comment 9: Please revise the section numbering (Section 6.6 is missing).
Response 9: Thank you for your comment. The section numbering has been revised, and the entire manuscript has been checked for similar errors.

Reviewer 2 Report
Comments and Suggestions for Authors
This paper offers a comprehensive exploration of brain tumor detection using deep learning, addressing a highly relevant and impactful topic. Below are several suggestions:
The abstract is overly dense with information. Simplify it to emphasize the main findings and key contributions of the paper.
Avoid repetition of general concepts such as the causes of brain tumors and their symptoms. Instead, streamline the introduction to focus on the significance and transformative potential of deep learning for diagnostic challenges in medical imaging.
Provide additional non-technical context to make the paper more accessible to a general audience. For example, in Figure 6, offer explanations of how the four extracted feature patterns differ and their implications for tumor classification.
Many paragraphs are overly long and dense, which hinders readability. Please break these into shorter ones.
Provide a more critical analysis of gaps in existing studies rather than listing datasets and methods. For example, elaborate on why certain advanced methods are underutilized.
Author Response
Comment 1: The abstract is overly dense with information. Simplify it to emphasize the main findings and key contributions of the paper.
Response 1: Thank you for your feedback. The abstract has been revised to simplify the content, emphasizing the main findings and key contributions of the paper.
Comment 2: Avoid repetition of general concepts such as the causes of brain tumors and their symptoms. Instead, streamline the introduction to focus on the significance and transformative potential of deep learning for diagnostic challenges in medical imaging.
Response 2: Thank you for your feedback. The paragraphs have been shortened as requested to avoid repetition of general concepts such as the causes of brain tumors and their symptoms. The introduction now focuses more on the significance and transformative potential of deep learning for addressing diagnostic challenges in medical imaging.
Comment 3: Provide additional non-technical context to make the paper more accessible to a general audience. For example, in Figure 6, offer explanations of how the four extracted feature patterns differ and their implications for tumor classification.
Response 3: Thank you for your comment. The description has been updated as requested, with additional non-technical context to make it more accessible and to explain the differences in the extracted feature patterns and their implications for tumor classification.
Comment 4: Many paragraphs are overly long and dense, which hinders readability. Please break these into shorter ones.
Response 4: Thank you for your valuable feedback. The entire text has been reviewed, and long paragraphs have been separated into smaller ones. Please check the entire text, where the beginning of each newly separated paragraph is highlighted in red. Additionally, the following statement has been added to Section 1.3, after Figure 4, in response to your comment.
“In this architecture of the current review paper, it is acknowledged that the long paragraphs discussing the papers may be difficult to read. To address this, we have made efforts to improve readability by introducing subsections. However, we believe the current subsectioning is the most suitable in this case, as it allows for clear presentation of the review results in tables, followed by bar charts. Furthermore, the review is structured around three major axes: classification, extraction, and segmentation. We believe that the inclusion of bar charts and diagrams will greatly assist readers in understanding the recent trends in each subfield and provide valuable insights for drawing meaningful conclusions. We trust that this organization will enhance the clarity of the manuscript while ensuring a comprehensive review of the topic.”
Comment 5: Provide a more critical analysis of gaps in existing studies rather than listing datasets and methods. For example, elaborate on why certain advanced methods are underutilized.
Response 5: Thank you for your comment. The introductory paragraph in Section 6 has been updated with the following revisions, ensuring that the gaps in research are clearly illustrated and critically analyzed for all obtained results of reviewed papers.
“The state-of-the-art in brain tumor detection using deep learning highlights significant advancements in methodologies, learning paradigms, and datasets, yet key gaps remain in fully leveraging these techniques. While CNNs and transfer learning have become dominant due to their efficiency in feature extraction, tumor segmentation, and classification, their reliance on large, labeled datasets still limits their applicability in diverse real-world settings. A critical gap exists in the generalizability of these models, particularly across different tumor types and patient populations. This issue is compounded by class overlapping, where tumor types exhibit similar features, leading to challenges in distinguishing them accurately. Moreover, while hybrid approaches have gained traction by combining various deep learning techniques for enhanced performance, these methods often lack the adaptability needed to address the complex, dynamic nature of brain tumor imaging. Custom architectures are promising but are frequently underutilized due to the challenge of tailoring solutions for specific datasets or tumor types.
Techniques such as Autoencoders, GANs, and LSTM networks remain less prevalent, despite their potential for addressing unique challenges such as data contextualization, temporal data modeling, and feature refinement. One reason for their underutilization could be the lack of sufficient research focused on their integration with brain tumor detection tasks. These methods could offer solutions to the complexities associated with imaging, such as noise reduction and the handling of varying tumor growth patterns over time. However, they often require more intricate setups and specialized knowledge to implement, which may contribute to their limited use.
Furthermore, datasets like BraTS, Kaggle, and T1-WCEMRI, while comprehensive and standardized, may not fully capture the variability seen in real clinical settings, where imaging techniques, tumor characteristics, and patient demographics can differ significantly. Personal clinical data presents an exciting opportunity to bridge this gap, yet the challenges of data privacy, annotation quality, and clinical integration hinder its widespread adoption. Additionally, while supervised learning continues to dominate, there is a need for more research into unsupervised or semi-supervised methods, particularly as these could provide solutions in cases where labeled data is scarce. Finally, while emerging techniques such as ViT, federated learning, and adversarial training represent the field’s drive toward innovation, these advanced methods are often underutilized due to their complexity, computational demand, and the need for specialized infrastructure, further limiting their adoption in mainstream clinical applications.
Despite these hurdles, the continued exploration of these underutilized methods and the resolution of existing gaps offer significant opportunities for advancing brain tumor detection and classification. To drive progress in this field, we propose several future directions categorized into five key areas, as illustrated in the diagram presented in Figure 13.”

Reviewer 3 Report
Comments and Suggestions for Authors
The manuscript presents a review of state-of-the-art deep learning techniques for brain tumor detection. Brain tumor detection is critical in medical research, and the application of deep learning in the field is a hot research topic. The review would be a great contribution to the community.
This is an overall well-written review with a comprehensive literature review, a clear description of existing research, and insightful summaries and proposals for future directions.
The review is in an almost ready-to-publish status. I recommend publication with minor revisions:
1. Figure 4: The curved arrows in the center are difficult to read. Please simplify or use more distinctive colors.
2. Figure 9: It needs a variable name alongside the color bar.
3. Figure 12 (b): The plot title should be Learning Paradigms.
4. Can you summarize the deep learning methods' accuracy or other evaluation metrics in a table, rather than in the text (lines 1206 - 1429), which is too lengthy and overwhelming to read.
Author Response
Comment 1: Figure 4: The curved arrows in the center are difficult to read. Please simplify or use more distinctive colors.
Response 1: Thank you for your feedback. Figure 4 has been updated to clearly show the direction of the arrows and enhanced with distinctive colors to improve readability. The updated figure also uses colors to help observe and differentiate the sections of the paper more effectively.
Comment 2: Figure 9: It needs a variable name alongside the color bar.
Response 2: Thank you for your comment. The label "Activation Intensity" has been added to the color bar in each subplot to clarify the information represented, addressing your feedback.
Comment 3: Figure 12 (b): The plot title should be Learning Paradigms.
Response 3: Thank you for your comment. The figure has been updated, and the plot title for Figure 12 (b) is now correctly labeled as "Learning Paradigms."
Comment 4: Can you summarize the deep learning methods' accuracy or other evaluation metrics in a table, rather than in the text (lines 1206 - 1429), which is too lengthy and overwhelming to read.
Response 4: Thank you for your suggestion regarding summarizing deep learning methods' accuracy or evaluation metrics in a table. While we appreciate the merit of such a summary for clarity, implementing this change across the entire paper would significantly alter its structure and exceed its intended scope. Our review focuses primarily on current methods, paradigms, and datasets rather than on comparative accuracy across varying approaches. Given the diverse datasets, varying sizes, and different experimental setups involved, summarizing accuracies alone would not provide a fair or comprehensive comparison aligned with our perspective. However, recognizing the importance of this analysis for future research, we have added the following statement to the introduction section:
“Finally, this review explores methods, paradigms, and datasets for feature extraction, segmentation, and classification, leaving a detailed comparative analysis of performance metrics, such as accuracy, as a valuable direction for future studies to better assess methodological effectiveness.”
